# Metabolic coupling of ROS generation and antioxidant synthesis by the GABA shunt pathway in myeloid-like blood progenitor cells of *Drosophila*

Manisha Goyal[1,2,3], Sakshi Tiwari[1,4], Jagriti Arora[1,5], Bruce Cooper[6], Ramaswamy Subramanian[3,7]*, Tina Mukherjee[1]*

1 iBRIC-Department of Biotechnology-Institute for Stem Cell Science and Regenerative Medicine (iBRIC-DBT-inStem), Bengaluru, Karnataka, India, 2 The University of Trans-Disciplinary Health Sciences & Technology (TDU), Bengaluru, Karnataka, India, 3 Department of Biological Sciences, Purdue University, West Lafayette, Indiana, United States of America, 4 Manipal Academy of Higher Education, Manipal, Karnataka, India, 5 Regional Centre for Biotechnology, Faridabad, Haryana, India, 6 Bindley Biosciences Centre, Purdue University, West Lafayette, Indiana, United States of America, 7 Weldon School of Biomedical Engineering, Purdue University, West Lafayette, Indiana, United States of America

* subram68@purdue.edu (RS); tinam@instem.res.in (TM)

## Abstract

Redox balance is crucial for normal development of stem and progenitor cells that reside in oxidative environments. In this study, we explore the mechanisms of redox homeostasis in such niches and show that myeloid-like blood progenitor cells of the *Drosophila* larval lymph gland, that generate reactive oxygen species (ROS), moderate it developmentally by *de novo* synthesizing glutathione (GSH) to ensure redox balance. During lymph gland development, as the blood-progenitor cells oxidize pyruvate via the TCA cycle leading to the generation of ROS, GABA-shunt restricts pyruvate dehydrogenase (PDH) activity and consequently TCA cycle flux. This moderation enables a metabolic rerouting of TCA-derived oxaloacetate (OAA) to pyruvate via gluconeogenesis, which is necessary to sustain serine levels, the rate-limiting precursor for *de novo* GSH synthesis. Disruption of GABA metabolism causes metabolic imbalance, marked by excessive PDH activity and heightened TCA cycle flux. This results in reduced OAA availability, impaired gluconeogenic capacity, and insufficient serine/GSH production, ultimately leading to ROS dys-regulation. Overall, this study identifies a unique metabolic framework in blood progenitor cells, where the GABA shunt, by restraining PDH and TCA cycle activity, maintains ROS at developmental levels. By coupling TCA-derived metabolites to GSH production, this state enables the TCA cycle to support both ROS generation and ROS scavenging, ensuring the developmental roles of ROS while preserving progenitor homeostasis.

**Data availability statement:** All relevant data are within the manuscript and its Supporting Information files.

**Funding:** o This study was funded by the Department of Science and Technology, Ministry of Science and Technology, India, CRG (Grant number CRG/2021/002815), by the Université de Strasbourg USIAS Indo French grant, and by The Wellcome Trust DBT India Alliance (Grant number IA/S/22/1/506259) awarded to T.M. The funders had no role in study design, data collection and analysis, decision to publish, or preparation of the manuscript.

**Competing interests:** The authors have declared that no competing interests exist.

## Author summary

Maintenance of fine redox balance is important for the homeostatic development of stem and early blood progenitors. GABA catabolism regulates redox homeostasis in *Drosophila* blood-progenitors by controlling TCA cycle activity to moderate ROS generation and promote ROS scavenging. This dual control of GABA in maintaining ROS homeostasis is important for proper development and differentiation of blood-progenitor cells. Diverse metabolic pathways are interconnected and regulated by GABA metabolism to facilitate ROS balance and homeostatic development of lymph glands. Specifically, for ROS scavenging, GABA catabolism co-ordinate pyruvate entry and OAA exit from TCA cycle to channel the flux towards GSH generation via gluconeogenic route through PEPCK2 resulting in serine formation. Serine generation regulation by GABA catabolic pathway act as a key step to accommodate GSH generation. Overall, the current work shows that GABA metabolism facilitates a metabolic environment for maintenance of ROS balance and blood-progenitor development.

## Introduction

Maintaining elevated reactive oxygen species (ROS) is a critical feature of many stem and progenitor cells [1,2]. While the developmentally generated ROS is necessary for coordinating their normal homeostatic functions [3], the mechanisms by which cells thriving in oxidative environments prevent the accumulation of oxidative stress remains an active area of research.

Specifically, blood stem and progenitor cells of common myeloid origin rely on mitochondrially generated ROS for their development [4]. However, any processes that excessively increases mitochondrial oxidation or disrupts mitochondrial function leading to aberrant ROS generation can impair progenitor maintenance and result in differentiation defects [5,6]. Therefore, it is crucial to maintain ROS levels within a specific threshold that allows ROS to function as a signaling molecule rather than a stress component. In this regard, the role of ROS scavenging and antioxidant mechanisms, which restore balance when ROS levels are excessively high, becomes increasingly relevant [7–9]. These mechanisms include both enzymatic and non-enzymatic processes, with antioxidant enzymes such as peroxidase, catalase, and superoxide dismutase (SOD), along with non-enzymatic metabolites like ascorbate, glutathione (GSH) and vitamin A, and form an integral part of the ROS scavenging system [10]. The generation and scavenging of ROS is shaped by various cellular changes [11]. In this study, we examine how hematopoietic progenitor cells, which develop in a dynamic niche, and are influenced by a range of environmental influences [12–16], coordinate ROS production and scavenging processes, so as to maintain redox balance.

Lymph gland, the primary definitive hematopoietic organ of *Drosophila*, present during larval stages of development, is a multilobed structure that houses multipotent

stem-like blood progenitor cells, which resemble vertebrate myeloid cells (reviewed in [17]. These progenitors reside in the inner core region of the organ termed medullary zone (MZ) and differentiate into all mature blood cell types, which are then localized in the outer zone of the lymph gland termed cortical zone (CZ) [17,18]. Similar to common myeloid progenitor cells [19], lymph gland progenitors have been shown to maintain higher ROS compared to their surrounding differentiated cells. The developmentally generated ROS is essential for priming these cells toward differentiation cues [5]. Various signaling modalities including NF-kB [20,21], JNK [22], Notch [23] and cues of nutritional/metabolic in origin like fatty acids [24] have been shown to regulate blood progenitor development by modulating ROS levels in the lymph gland [2]. However, aberrant ROS levels impairs progenitor development and homeostasis, thus highlighting the importance of ROS regulation in progenitor development [5]. ROS generation in progenitor cells is linked to mitochondrial/TCA activity [5,25], and antioxidants such as catalase, SOD, and glutathione peroxidase (Gtpx) have been shown to play essential roles in maintaining ROS homeostasis [5]. These antioxidants help neutralize excess ROS, protecting progenitor cells from oxidative damage and ensuring proper development and function. Therefore, the balance between ROS production and antioxidant activity is essential for controlled differentiation and survival of these myeloid-like blood progenitor cells. The understanding of regulatory mechanisms that coordinate ROS generation with scavenging to maintain a delicate ROS balance in the lymph gland blood system forms the central question of the current study.

Our previous research identified the tricarboxylic acid (TCA) cycle as a source of ROS in the progenitor cells [25]. Moreover, we found that neuronally derived GABA (γ-aminobutyric acid), [13], its uptake and metabolism by the progenitor cells [26] control ROS production through limiting TCA cycle activity [25]. The study demonstrated that GABA taken up by progenitor cells via the GABA transporter (Gat) and metabolized through the GABA-shunt pathway, produces succinate, which regulates pyruvate dehydrogenase (PDH) activity and controls the rate of the TCA cycle (Fig 1A). Loss of GABA function led to increased TCA cycle activity and elevated ROS levels in the progenitor cells and highlighted a key redox modulatory role of GABA in the progenitor cells.

The findings from our previous work, led us to further investigate the redox modulatory effects of the GABA-shunt pathway. In this study, we describe its role in the generation of a key antioxidant, glutathione (GSH), by the blood-progenitor cells and establishment of their anti-oxidant potential. Glutathione serves as a substrate for enzymes like glutathione S-transferase, and Gtpx, which are essential for mitigating redox stress and preventing oxidative damage [27,28]. These enzymatic ROS scavengers have been shown to play a vital role in the maintenance of the lymph gland progenitor cells in their undifferentiated state [5]. Thus, the heightened reliance of progenitor cells on these enzymatic scavengers led us to explore GSH regulation and we observed that progenitor cells maintain elevated levels of GSH in them. The accumulation of GSH in progenitor cells in addition to elevated ROS also detected in them, suggested specialized developmental program that operated towards maintaining effective ROS scavenging mechanisms in the myeloid-like blood progenitor cells.

Glutathione is a tripeptide comprised of three amino acids (cysteine, glutamate, and glycine), and cycles between oxidized and reduced states [29]. The presence of total GSH in the progenitor cells alluded to its *de novo* generation. By employing a combination of immunohistochemical and mass spectrometry-based approaches, we investigated the underlying modality leading to GSH accumulation in progenitor cells. The investigations revealed an unexpected control of pyruvate metabolism in establishing GSH levels in them. Active pyruvate oxidation and TCA cycling generates ROS in progenitor cells, but the extent of pyruvate utilization and TCA cycle activity defines the cellular potential to generate serine. Serine is a key biosynthetic intermediate to generate GSH, and its levels become rate-limiting in defining progenitor GSH levels. Thus, to achieve this balance, GABA metabolism coordinates pyruvate utilization in the progenitor cells whereby it allows controlled pyruvate entry into the TCA and simultaneously enables a state that re-routes pyruvate for serine synthesis. We show that GABA metabolism diverts pyruvate to generate serine via gluconeogenesis, and this ascertains a metabolic program that allows progenitor cells to develop homeostatically, producing ROS while simultaneously scavenging it. The use of GABA metabolism by the blood progenitor cells to counter redox stress at the nexus of regulating pyruvate metabolism is a central finding from the study. The reliance of vertebrate hematopoietic stem and

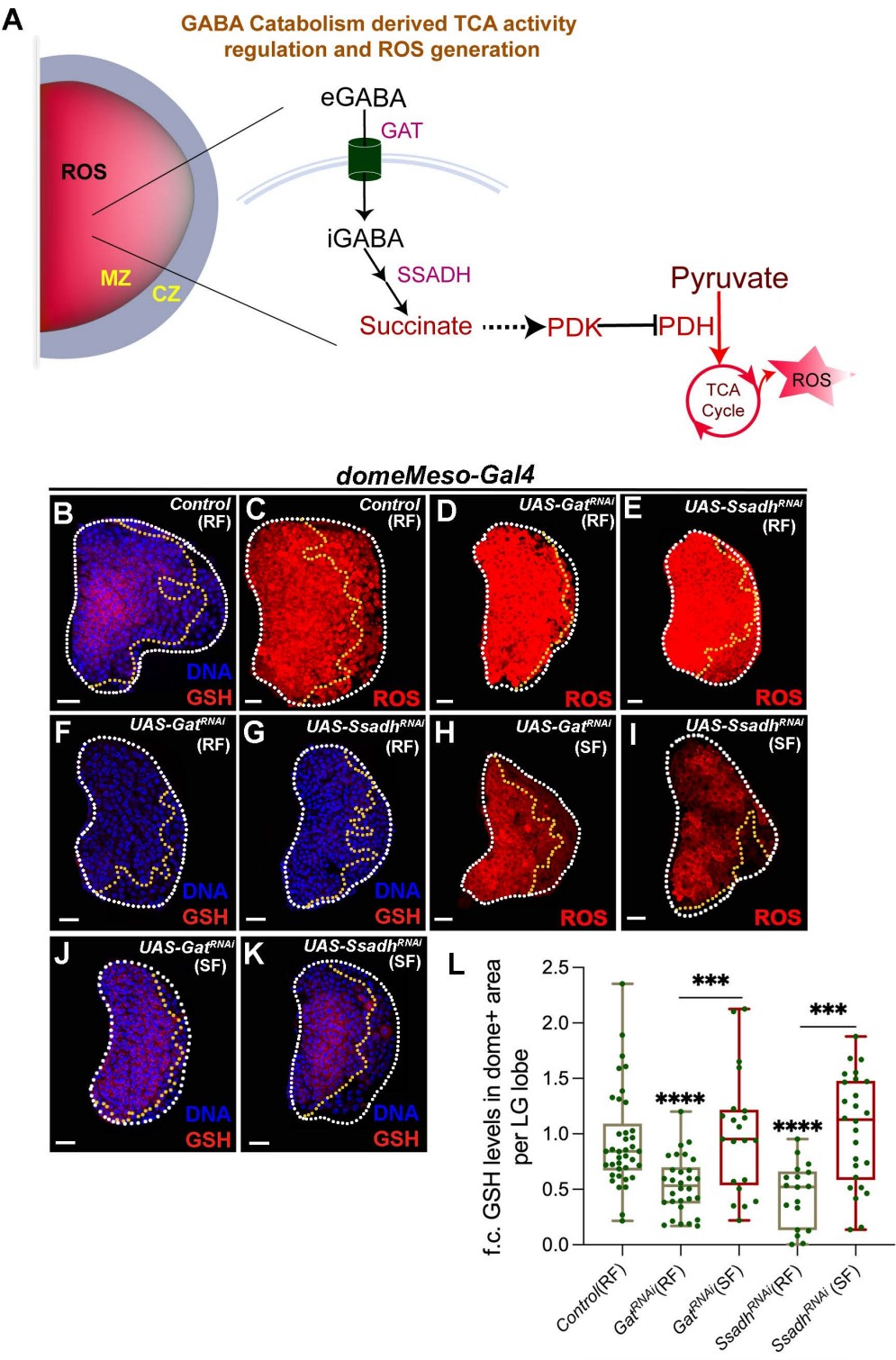

**Fig 1. GABA catabolic pathway in *Drosophila* blood-progenitor cells control their GSH levels.** RF is regular food; SF is succinate food. Data is presented as median plots (***p < 0.001 and ****p < 0.0001). Mann-Whitney test is applied for **L**. In **L**, 'n' is total number of lymph gland lobes analysed and is represented by a green dot. Scale bar: 20μm. DNA is stained with DAPI in blue. Comparisons for significance are done with control and

PLOS Genetics

with respective genetic conditions for rescue combinations (red bars), which are indicated by horizontal lines drawn above the box plots. White border demarcates the lymph gland lobe and yellow border marks the dome positive area towards the left side. The respective lymph gland images with dome+ (green) are shown in S1 Fig. **(A)** Schematic representation of GABA catabolism and its regulation of TCA cycle activity. In blood-progenitor cells (MZ), which maintain elevated levels of ROS, extracellular GABA (eGABA) is internalized by GAT and catabolized into succinate by a two-step process including second, rate-limiting step catalysed by SSADH. GABA catabolism derived succinate regulates PDK activity (pPDK) to control PDH phosphorylation, which restricts TCA cycle activity and generation of ROS. CZ-cortical zone, eGABA-extracellular GABA, GABA-γ-aminobutyric acid, GAT-GABA Transporter, iGABA-intracellular GABA, MZ-medullary zone, PDH-pyruvate dehydrogenase, PDK-pyruvate dehydrogenase kinase, ROS-reactive oxygen species, SSADH-succinic semialdehyde dehydrogenase, TCA cycle-tricarboxylic acid cycle. **(B-K)** Representative images showing ROS (red) and GSH (red) levels in the lymph gland progenitor cells (area marked within the yellow dotted line) from different genetic backgrounds. Control (*domeMeso-Gal4,UAS-GFP/+*) showing **(B)** GSH levels, which is elevated in the progenitor cells and **(C)** elevated ROS (stained with DHE), in them. Progenitor specific expression of **(D, F)** *Gat^RNAi^* (*domeMeso-Gal4,UAS-GFP;UAS-Gat^RNAi^*) and **(E, G)** *Ssadh^RNAi^* (*domeMeso-Gal4,UAS-GFP;UAS-Ssadh^RNAi^*) leads to an increase in **(D, E)** ROS levels and reduction in **(F, G)** GSH levels. For comparison, refer to control ROS in **(C)** and control GSH **(B)**. **(H-K)** Succinate supplementation to **(H, J)** *Gat^RNAi^* (SF, *domeMeso-Gal4,UAS-GFP;UAS-Gat^RNAi^*) and **(I, K)** *Ssadh^RNAi^* (SF, *domeMeso-Gal4,UAS-GFP;UAS-Ssadh^RNAi^*), rescues blood-progenitor **(H, I)** ROS and **(J, K)** GSH levels respectively. For comparison, refer to **(D, F)** *Gat^RNAi^* and **(E, G)** *Ssadh^RNAi^*. For GSH quantifications, refer to **L**. **(L)** Quantification of blood-progenitor GSH levels (fold change, f.c.) in *domeMeso>GFP/+* (control, n = 38)*, domeMeso>GFP/Gat^RNAi^* (RF, n = 30, p < 0.0001), *domeMeso>GFP/Gat^RNAi^* (SF, n = 21, p = 0.0009, in comparison to *Gat^RNAi^*), *domeMeso>GFP/Ssadh^RNAi^* (RF, n = 18, p < 0.0001) and *domeMeso>GFP/ Ssadh^RNAi^* (SF, n = 26, p = 0.0001, in comparison to *Ssadh^RNAi^*).

progenitor cells on TCA/mitochondria to generate ROS [30–33] and their dependency on GABA is much recently shown [34]. Therefore, we speculate that the metabolic program identified in this study may well be conserved in the mammalian hematopoietic niche and also across other stem and progenitor cells residing in oxidative niches [19,35–37].

## Results

### Control of blood progenitor glutathione levels by the GABA-shunt pathway

The non-enzymatic antioxidant glutathione (GSH) was analysed in *Drosophila* third instar larval lymph glands using a specific antibody generated against it (Abcam 9443, [38]. The data revealed that GSH levels were significantly higher in the progenitor cells in contrast to the differentiated cells of the cortical zone (CZ) (Figs 1B, S1A, and S1A"). Notably, the antibody detects total glutathione and does not distinguish between its oxidized and reduced forms [38]. As GSH is a small metabolite, its detection like that of other small metabolites such as amino acids [13,26,39], is sensitive to washing protocols (see methods for details). Under stringent wash conditions, GSH detection is lower than (S1A-A" Fig) seen with milder detergent washes, which preserved higher GSH signal in the lymph glands (S1B-B" Fig). Throughout the study, we used stringent washes to enable detection of metabolites in the lymph gland.

To understand the dynamics of GSH levels and for the specificity of the antibody against total GSH, we reared control larvae from early first instar developmental stage on food supplemented with 0.1% GSH and analysed these lymph glands for their GSH levels. Interestingly, wandering 3rd instar larvae obtained from this condition showed increased progenitor GSH levels (S1C-D Fig), and implied the antibody detected total GSH and is sensitive to any change in GSH levels in the progenitor compartment. The elevated GSH detected in progenitor cells was also consistent with enzymatic antioxidant activities linked to glutathione reported in the literature [5,20].

We have previously demonstrated that GABA metabolism is crucial for moderating progenitor TCA cycle activity and plays an important role in regulating ROS generation. Furthermore, ROS moderation through GABA catabolism via controlling TCA cycle activity is central to regulate homeostatic lymph gland growth [25]. We found that down-regulating components of the GABA catabolic pathway, not only affected progenitor ROS levels (Figs 1C-E and S1E-G) but also led to reduction in their GSH levels (Figs 1F, 1G, 1L and S1H-J'). Specifically, blocking progenitor GABA uptake, by knocking down the transporter, *Gat* (Figs 1F, S1I, and S1I') or downregulating the rate-limiting GABA-shunt catabolic enzyme, *Ssadh* (Figs 1G and S1J, S1J'), using the progenitor-specific driver *domeMeso-Gal4,* resulted in a significant reduction in their GSH levels. Supplementing succinate, the by-product of GABA breakdown, which restores progenitor ROS levels (Figs 1H, 1I and S1K, S1L) in conditions where *Gat* or *Ssadh* RNA*i* is expressed, also significantly increased progenitor

GSH levels (Figs 1J, 1K, 1L and S1M-N'). These data suggested an unexpected role for GABA shunt pathway in progenitor GSH homeostasis alongside its known role in moderating ROS generation that we have previously reported [25]. The mechanism by which GABA influenced GSH levels, was therefore explored.

## Regulation of glutathione synthesis by GABA catabolic pathway

The antibody used to detect GSH does not distinguish between its oxidized and reduced forms; therefore, the measured GSH levels in progenitor cells represented its total concentration [38]. The observed reduction in GSH levels in *Gat^RNAi* and *Ssadh^RNAi*, followed by recovery upon succinate supplementation (Fig 1), led us to hypothesize a defect in GSH synthesis, as opposed to any changes in its oxidized or reduced state. This prompted us to explore the mechanisms regulating GSH synthesis and the involvement of GABA metabolism in this process.

Glutathione (GSH) is a non-protein thiol metabolite synthesized from the amino acid cysteine, glutamate, and glycine (Fig 2A). The rate-limiting step in GSH synthesis involves the combination of cysteine and glutamate to form γ-glutamyl cysteine. In the final step, glycine is added, resulting in the formation of GSH (Fig 2A) [29]. To determine whether the GSH reduction observed in GABA catabolic knockdowns was due to changes in the intermediates required for GSH synthesis, we analyzed progenitor levels of these intermediates using specific antibodies, where available and via mass spectrometry (see Methods).

We found that glutamate levels upon disruption of the GABA transporter (*Gat*) or *Ssadh* in blood progenitor cells revealed no significant change both immunohistochemically with an anti-glutamate antibody (Figs 2B-D, 2T and S2A-C) and mass-spectrometrically (S2U Fig). However, the levels of cysteine when assessed using a cysteine-specific antibody (sc-69954), which marks cysteinylated proteins in the cells and is a readout for overall intra-cellular cysteine levels [40], showed a reduction in cysteine levels (Fig 2E-G and 2U). In control lymph glands, cysteine is detected uniformly across progenitor and differentiating cells (Figs 2E and S2D), however progenitor-specific loss of *Gat* or *Ssadh* led to a significant reduction in overall cysteine levels (Figs 2E-G, 2U and S2D-F). Despite RNA*i* being restricted to progenitors, the overall change in cysteine levels suggests a non-cell-autonomous regulation of cysteine metabolism within the lymph gland. It is possible that cysteine or its metabolic precursors may be synthesized predominantly in progenitor cells and subsequently shared with or influence the metabolic state of neighbouring differentiated cells as seen for other metabolites, such as adenosine [41].

Cysteine was also analyzed mass-spectrometrically, but we failed to detect any signal and speculate limited sample size and cysteine's inherent instability due to its thiol group [42] as the underlying factor. Glycine was assessed using mass-spectrometry and using this approach we observed no detectable changes for glycine levels in *Gat* and *Ssadh* loss of function condition (S2V Fig). These findings demonstrated that GABA catabolism most likely regulated progenitor cysteine levels, which consequently regulated their GSH synthesis. The reduction in GSH in these catabolic knockdowns could therefore arise as a consequence of reduced cysteine levels.

Therefore, to address if the lower amounts of cysteine led to reduction in GSH synthesis, we supplemented *domeMeso>Gat^RNAi* and *domeMeso>Ssadh^RNAi* conditions with N-acetylcysteine (NAC), an analog for cysteine. This supplementation restored progenitor GSH levels in *Gat^RNAi* and *Ssadh^RNAi* to nearly comparable levels seen in control lymph glands (Figs 2H-L, 2V and S2G-K). Importantly, cysteine is a sulfur amino acid and can be derived from methionine through its cycling and the transsulfuration pathway [43]. Therefore, we supplemented *Gat^RNAi* larvae with methionine, however, its supplementation failed to show any restoration of GSH levels (Figs 2M, 2V and S2L). This data indicated that NAC mediated recovery of GSH was specific to restoration of cysteine and not a non-specific rescue by any other sulfur amino acid. Furthermore, the data also suggested that the cysteine synthesis regulation by GABA catabolism was independent of its regulation of methionine. Therefore, we conducted the next set of analyses to investigate how GABA catabolism supported cysteine synthesis in progenitor cells.

We know that downstream of GABA catabolism, succinate derived from its breakdown restored GSH levels in the knockdown backgrounds (Fig 1L). If succinate supplementation in *domeMeso>Gat^RNAi* and *domeMeso>Ssadh^RNAi* larvae

**A**

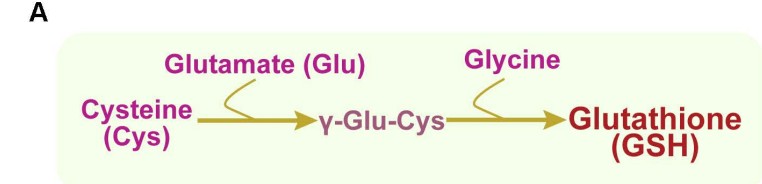

*domeMeso-Gal4*

[Fig 2 microscopy panels B–M, N–S]

*domeMeso-Gal4*

[Plots T, U, V]

**Fig 2. Regulation of glutathione (GSH) synthesis by GABA catabolic pathway.** RF is regular food, SF is succinate food, NAC is N-acetylcysteine and Met is methionine supplemented food. Data is presented as median plots (**p < 0.01; ***p < 0.001; ****p < 0.0001 and ns is non-significant). Mann-Whitney test is applied for **T-V**. In **T-V**, 'n' is total number of lymph gland lobes analysed and is represented by a green dot. Scale bar: 20μm. DNA

is stained with DAPI in blue. Comparisons for significance are done with control and with respective genetic conditions for rescue combinations (red bars), which are indicated by horizontal lines drawn above the box plots. White border demarcates the lymph gland lobe and yellow border marks the dome positive area towards the left side. The respective lymph gland images with dome+ (green) are shown in S2 Fig. **(A)** Schematic representation of glutathione (GSH) synthesis pathway. Cysteine (Cys) and glutamate (Glu) conjugate to form γ-glutamyl-cysteine, and further addition of glycine leads to GSH formation. **(B-D)** Representative images showing glutamate (Glu, red) levels in lymph gland progenitor cells across different genetic backgrounds. In comparison to **(B)** control (*domeMeso-Gal4,UAS-GFP*/+) lymph gland showing uniform glutamate levels across all cells of the tissue, including the progenitor-cells (area within the yellow dotted line), expressing **(C)** *Gat^RNAi* (*domeMeso-Gal4,UAS-GFP;UAS-Gat^RNAi*) and **(D)** *Ssadh^RNAi* (*domeMeso-Gal4,UAS-GFP;UAS-Ssadh^RNAi*) in the progenitor cells does not affect their glutamate levels. For quantifications, refer to **T**. **(E-G)** Representative images showing cysteine (Cys, red) levels in lymph gland progenitor cells (area marked within the yellow dotted line) from different genetic backgrounds. **(E)** Control (RF, *domeMeso-Gal4,UAS-GFP*/+) lymph gland showing relatively uniform cysteine levels in all cells of the lymph gland including progenitor-cells (area demarcated within the yellow border), expressing **(F)** *Gat^RNAi* (RF, *domeMeso-Gal4,UAS-GFP;UAS-Gat^RNAi*) and **(G)** *Ssadh^RNAi* (RF, *domeMeso-Gal4,UAS-GFP;UAS-Ssadh^RNAi*) in the progenitor cells leads to reduction in cysteine levels as compared to control **(E)**. For quantifications, refer to **U**. **(H-M)** Representative images showing GSH (red) levels in lymph gland progenitor cells (area marked within the yellow dotted line) from different genetic backgrounds. **(H)** Control (RF, *domeMeso-Gal4,UAS-GFP*/+) lymph gland showing GSH levels. While, expressing **(I)** *Gat^RNAi* (RF, *domeMeso-Gal4,UAS-GFP;UAS-Gat^RNAi*) and **(J)** *Ssadh^RNAi* (RF, *domeMeso-Gal4,UAS-GFP;UAS-Ssadh^RNAi*) in the progenitor cells leads to reduction in GSH levels, supplementing these genetic conditions with **(K, L)** N-acetylcysteine (NAC), restores GSH levels in **(K)** *Gat^RNAi* (NAC, *domeMeso-Gal4,UAS-GFP;UAS-Gat^RNAi*) and **(L)** *Ssadh^RNAi* (NAC, *domeMeso-Gal4,UAS-GFP;UAS-Ssadh^RNAi*) to levels almost comparable to control **(H)**. **(M)** Methionine supplementation to *Gat^RNAi* (Met, *domeMeso-Gal4,UAS-GFP;UAS-Gat^RNAi*) does not recover GSH levels. For comparison, also refer to *Gat^RNAi* **(I)** and *Ssadh^RNAi* **(J)** raised on regular food (RF). For quantifications, refer to **V**. **(N-O)** Succinate supplementation to **(N)** *Gat^RNAi* (SF, *domeMeso-Gal4,UAS-GFP;UAS-Gat^RNAi*) and **(O)** *Ssadh^RNAi* (SF, *domeMeso-Gal4,UAS-GFP;UAS-Ssadh^RNAi*) restores blood-progenitor cysteine levels. For comparison, refer to **(F)** *Gat^RNAi* and **(G)** *Ssadh^RNAi* animals raised on regular food (RF) and control **(E)**. For quantifications, refer to **U**. **(P-Q)** Progenitor specific expression of *Pdha^RNAi* in *Gat^RNAi* animals (*domeMeso-Gal4,UAS-GFP; UAS-Pdha^RNAi;UAS-Gat^RNAi*) leads to a recovery of blood-progenitor **(P)** cysteine and **(Q)** GSH levels. Compare with **(F)** Cys and **(I)** GSH in *Gat^RNAi* (*domeMeso-Gal4,UAS-GFP;UAS-Gat^RNAi*). For quantifications, refer to S2V and S2W Fig. **(R, S)** Expressing **(R)** *Pdk^RNAi* (*domeMeso-Gal4,UAS-GFP;UAS-Pdk^RNAi*) and **(S)** *Pdha^RNAi* (*domeMeso-Gal4,UAS-GFP;UAS-Pdha^RNAi*) in the progenitor cells, does not reveal any dramatic change in GSH levels. Compare to control **(H)**. For quantifications, refer to S2W Fig. **(T)** Quantification of blood-progenitor glutamate (Glu) levels (fold change, f.c.) in *domeMeso>GFP*/+ (control, n = 47), *domeMeso>GFP/Gat^RNAi* (n = 42, p = 0.1570) and *domeMeso>GFP/Ssadh^RNAi* (n = 19, p = 0.5180). **(U)** Quantification of blood-progenitor cysteine (Cys) levels (fold change, f.c.) in *domeMeso>GFP*/+ (control, RF, n = 36), *domeMeso>GFP/Gat^RNAi* (RF, n = 38, p < 0.0001), *domeMeso>GFP/Gat^RNAi* (SF, n = 41, p < 0.0001 in comparison to *Gat^RNAi*, RF), *domeMeso>GFP/Ssadh^RNAi* (RF, n = 50, p < 0.0001), and *domeMeso>GFP/Ssadh^RNAi* (SF, n = 18, p = 0.0075 in comparison to *Ssadh^RNAi*, RF). **(V)** Quantification of blood-progenitor GSH levels (fold change, f.c.) in *domeMeso>GFP*/+ (control, RF, n = 33), *domeMeso>GFP/Gat^RNAi* (RF, n = 37, p < 0.0001), *domeMeso>GFP/Gat^RNAi* (NAC, n = 32, p < 0.0001 in comparison to *Gat^RNAi*, RF), *domeMeso>GFP/Gat^RNAi* (Methionine, n = 22, p = 0.0012 in comparison to *Gat^RNAi*, RF), *domeMeso>GFP/Ssadh^RNAi* (RF, n = 12, p < 0.0001), and *domeMeso>GFP/Ssadh^RNAi* (NAC, n = 23, p = 0.0004 in comparison to *Ssadh^RNAi*, RF).

effectively restored cysteine levels was therefore asked. Interestingly, we observed that succinate restoration could re-establish cysteine levels in the lymph glands (Figs 2N, 2O, 2U and S2M, S2N). These results suggested that GABA catabolism derived succinate controls progenitor cysteine levels.

## Pyruvate metabolism and its control on progenitor GSH generation

Our previous work has shown that, GABA-shunt derived succinate moderates PDH activity to control the entry of pyruvate into the TCA cycle, resulting in restricted TCA rate and consequently ROS generation [25]. If elevated PDH activity in loss of GABA metabolism contributed to any lowering of GSH synthesis was therefore explored. For this, we down-regulated *Pdha* in *Gat^RNAi* condition and assessed for cysteine and GSH levels. We found that this genetic condition restored lymph cysteine levels, almost comparable to that seen in control lymph glands (Figs 2P and S2O, S2W) and very importantly progenitor GSH levels were restored as well (Figs 2Q and S2R, S2X). The data implicated elevated PDH activity in absence of GABA metabolism as the underlying cause for dampened blood progenitor GSH production as well. The regulation of pyruvate oxidation by PDH emerged as a necessary step not only for maintaining TCA cycle rate but also for supporting the antioxidant potential by driving cysteine and consequently GSH synthesis in the progenitor cells.

Therefore, to address the influence of pyruvate oxidation in progenitor GSH levels and if this was purely maintained as a consequence of PDH activity, we increased PDH activity independent of altering GABA catabolism in the progenitor cells. We undertook this by downregulating *Pdk* (Pyruvate dehydrogenase kinase), the kinase that phosphorylates PDH and inactivates it [44,45]. *Pdk* downregulation in progenitor cells did not affect their cysteine (S2P, S2P' and S2W Fig) or

GSH (Figs 2R and S2S, S2X) levels. The data was intriguing and showed that heightened pyruvate oxidation via increasing PDH activity was insufficient to impair GSH generation, even though this genetic condition increases progenitor ROS levels [25]. Contrarily, we also blocked pyruvate oxidation by down-regulating *Pdha* in blood-progenitors and this condition as well did not reveal any major difference in progenitor cysteine (S2Q, S2Q' and S2W Fig) or GSH (Figs 2S and S2T, S2X) levels.

These differences were not due to *Gal4* dilution, as RNA*i* efficiency analysis confirmed comparable reductions in PDH (S3A-C' and S3M Fig) and Gat (S3D-F' and S3N Fig) protein levels in *Pdha^RNAi;Gat^RNAi* co-expression compared to single knockdowns. Similarly, progenitor-specific *Pdk* and *Pdha* knockdowns were validated by assessing PDH and phosphorylated PDH (pPDH) levels: PDH marks active enzyme, whereas pPDH, phosphorylation mediated by PDK, marks inactive PDH. *Pdha^RNAi* reduced both PDH and pPDH, while *Pdk^RNAi* reduced only pPDH, confirming efficient progenitor-specific loss using *domeMeso-Gal4* (S3G-L Fig); [25].

Collectively, these results indicate that modulating PDH activity alone does not impact GSH synthesis. In the presence of GABA catabolism, additional pyruvate metabolism pathways likely maintain Cys/GSH synthesis. However, in its absence, excessive PDH activity may limit pyruvate availability for these pathways, thereby impairing Cys/GSH synthesis. Under such conditions, *Pdha* knockdown in *Gat^RNAi* progenitors likely restores pyruvate availability, rescuing cysteine and GSH synthesis. Together, these findings identify pyruvate availability, rather than PDH activity per se, as a key determinant of progenitor cysteine and GSH levels.

## GABA catabolism controls progenitor GSH generation via regulating de novo serine synthesis

Pyruvate is an essential cellular metabolite and plays not just an important role in driving the TCA cycle [46] but also a variety of other intracellular processes, like glycolysis, gluconeogenesis and fatty acid synthesis [47]. In this regard, the importance of GABA shunt and its regulatory role on influencing pyruvate metabolism to systematically enable cysteine synthesis was proposed.

Cysteine is a non-essential amino acid and is synthesized in cells via the conserved transsulfuration pathway (TSP) (Fig 3A). Methionine and serine, the precursor amino acids in TSP, combines through a series of steps to gives rise to cysteine [43]. While methionine supplementation data shown above (Fig 2M) ruled out any contribution of it in GSH synthesis, any limitation in serine levels culminating in reduced cysteine generation was investigated. Serine is a component of TSP and when analysed for total serine in *Gat^RNAi* conditions, we observed a significant decrease in total lymph gland serine levels (Fig 3B). If the reduction in serine contributed towards cysteine and GSH generation in *Gat^RNAi* condition was next assessed. We undertook supplementation of serine in *Gat^RNAi* condition and observed a significant rescue in both cysteine (Figs 3C-E, 3L and S4A-C) and GSH (Figs 3F-H, 3M and S4D-F) levels. Moreover, supplementing serine also restored the increased ROS phenotype detected in *Gat^RNAi* (Figs 3I-K, 3N and S4G-I) condition to moderate levels. Consistent with lymph gland growth inhibitory effect of heightened ROS production [25], the small sized lymph gland phenotype in *Gat^RNAi* condition was also restored by serine supplementation to sizes almost comparable to control lymph glands (Fig 3K and 3O). These data implied that GABA catabolism in lymph gland blood progenitor cells controlled their serine levels, whose availability, through GSH generation controlled progenitor ROS scavenging. The dual control exerted by GABA catabolism on ROS production and progenitor ROS scavenging overall controlled homeostatic lymph gland growth and development.

In this context, intracellular availability of serine and any correlation with pyruvate metabolism was explored. Serine levels can be managed either via its uptake through transporters [48] or through *de novo* routes that generate it [49]. *de novo* synthesis becomes interesting, because pyruvate via gluconeogenesis can be a predominant source to generate serine [50]. It is therefore possible that increased pyruvate oxidation in *Gat^RNAi* impaired pyruvate availability for gluconeogenesis and consequently serine biosynthesis. To understand this, we undertook a $U^{13}C$-pyruvate isotope based metabolic labelling analysis in wandering 3rd instar control and *Gat^RNAi* lymph glands and assessed for pyruvate metabolism.

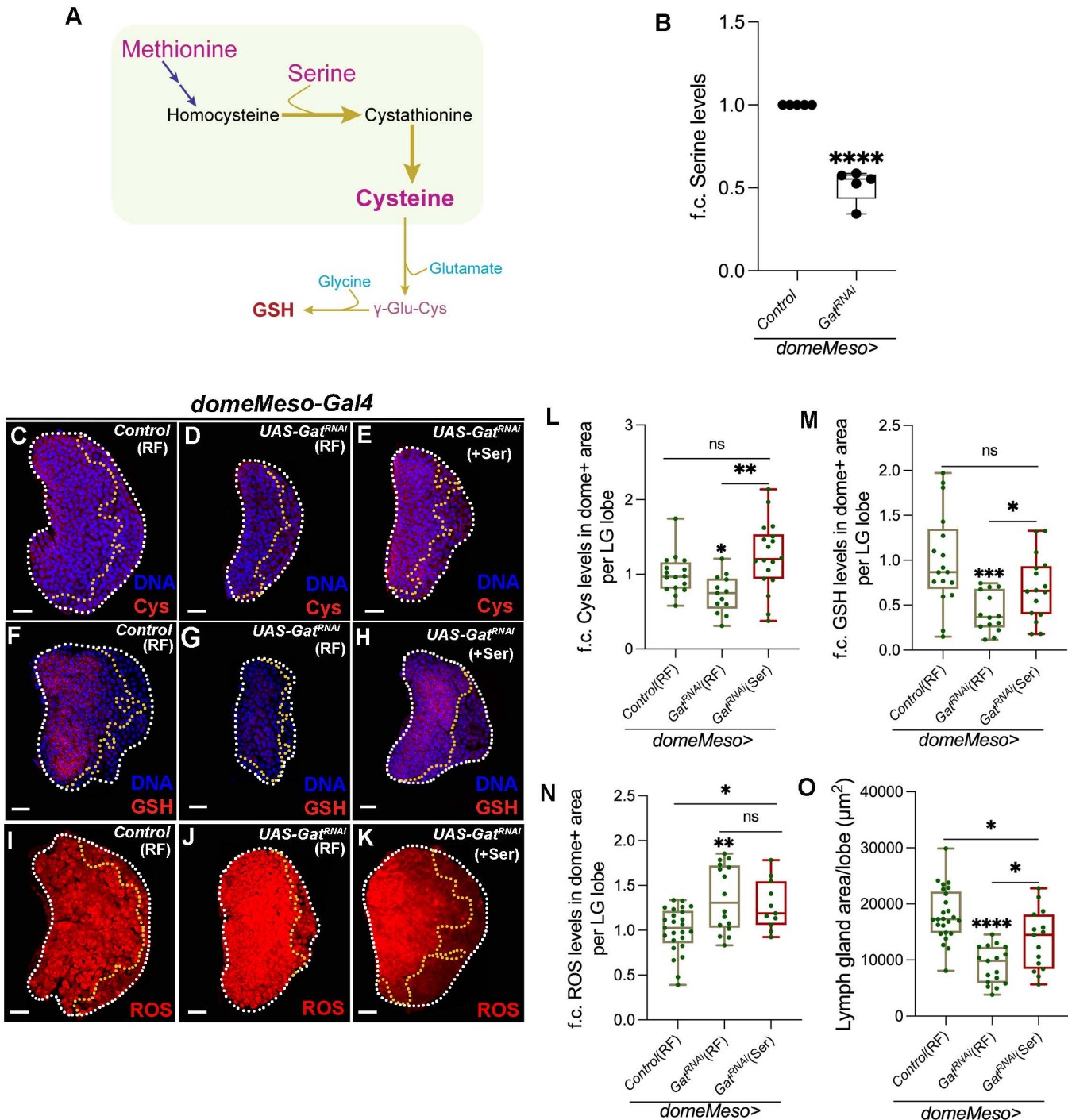

**Fig 3. GABA catabolism controls progenitor GSH generation via regulating *de novo* serine synthesis.** RF is regular food, and Ser is serine supplemented food. Data is presented as median plots (*p<0.05; **p<0.01; ***p<0.001; ****p<0.0001 and ns is non-significant). t-test is applied for **B** and Mann-Whitney test is applied for **L-O**. In **B** 'n' is sample size and 'N' is number of experimental repeats and shown by black dot. In **L-O**, 'n' is total number of lymph gland lobes analysed and is represented by a green dot. Scale bar: 20μm. DNA is stained with DAPI in blue. Comparisons for significance are done with control and with respective genetic conditions for rescue combinations (red bars), which are indicated by horizontal lines drawn above the box plots. White border demarcates the lymph gland lobe and yellow border marks the dome positive area towards the left side. The respective lymph gland images with dome⁺ (green) are shown in S4 Fig. **(A)** Schematic representation of transsulfuration pathway (shown in the green box) and glutathione (GSH) synthesis. Methionine and serine combine to form cysteine via the transsulfuration pathway (TSP). Sequential addition of

glutamate and glycine to cysteine form GSH. **(B)** Relative steady state levels (fold change, f.c.) of serine in *domeMeso>GFP/+* (control, N = 5, n = 15) *and domeMeso>GFP/Gat^RNAi* (N = 5, n = 18, p < 0.0001), where *Gat^RNAi* in the progenitor cells leads to reduction in lymph gland serine levels as compared to control. **(C-E)** Representative images showing cysteine (Cys, red) levels in lymph gland progenitor cells (area marked within the yellow dotted line) from different genetic backgrounds. **(C)** Control (RF, *domeMeso-Gal4,UAS-GFP/+*) lymph gland showing relatively uniform cysteine levels in all cells of the lymph gland including progenitor-cells. While, expressing **(D)** *Gat^RNAi* (RF, *domeMeso-Gal4,UAS-GFP;UAS-Gat^RNAi*) in the progenitor cells leads to reduction in cysteine levels as compared to control **(C)**, supplementing this genetic condition with **(E)** serine (Ser, *domeMeso-Gal4,UAS-GFP;UAS-Gat^RNAi*) recovers cysteine. For comparison, refer to **(D)** *Gat^RNAi* raised on regular food (RF). For quantifications, refer to **L. (F-H)** Representative images showing GSH (red) levels in lymph gland progenitor cells (area marked within the yellow dotted line) from different genetic backgrounds. **(F)** Control (RF, *domeMeso-Gal4,UAS-GFP/+*) lymph gland showing GSH levels in lymph gland progenitor-cells. While, expressing **(G)** *Gat^RNAi* (RF, *domeMeso-Gal4,UAS-GFP;UAS-Gat^RNAi*) in the progenitor cells leads to reduction in GSH levels as compared to control **(F)**, supplementing this genetic condition with **(H)** serine (Ser, *domeMeso-Gal4,UAS-GFP;UAS-Gat^RNAi*) recovers GSH levels. For comparison, refer to **(G)** *Gat^RNAi* raised on regular food (RF). For quantifications, refer to **M. (I-K)** Representative images showing ROS levels (red, area marked within the yellow dotted line) from different genetic backgrounds, **(I)** control (RF, *domeMeso-Gal4,UAS-GFP/+*) lymph gland showing ROS levels in lymph gland progenitor-cells. While, expressing **(J)** *Gat^RNAi* (RF, *domeMeso-Gal4,UAS-GFP;UAS-Gat^RNAi*) leads to increase in progenitor ROS as compared to control **(I)**, supplementing this genetic condition with **(K)** serine (Ser, *domeMeso-Gal4,UAS-GFP;UAS-Gat^RNAi*) recovers the increased ROS levels. For comparison, refer to **(J)** *Gat^RNAi* raised on regular food (RF). For quantifications, refer to **N. (L)** Quantification of blood-progenitor cysteine (Cys) levels (fold change, f.c.) in *domeMeso>GFP/+* (control, RF, n = 16), *domeMeso>GFP/Gat^RNAi* (RF, n = 13, p = 0.0132), and *domeMeso>GFP/Gat^RNAi* (Ser, n = 18, p = 0.0754 in comparison to control and p = 0.0017 in comparison to *Gat^RNAi*, RF). **(M)** Quantification of blood-progenitor GSH levels (fold change, f.c.) in *domeMeso>GFP/+* (control, RF, n = 17), *domeMeso>GFP/Gat^RNAi* (RF, n = 14, p = 0.0003), and *domeMeso>GFP/Gat^RNAi* (Ser, n = 16, p = 0.0942 in comparison to control and p = 0.0308 in comparison to *Gat^RNAi*, RF). **(N)** Quantification of blood-progenitor ROS levels (fold change, f.c.) in *domeMeso>GFP/+* (control, RF, n = 25), *domeMeso>GFP/Gat^RNAi* (RF, n = 16, p = 0.0088), and *domeMeso>GFP/Gat^RNAi* (Ser, n = 11, p = 0.0161 in comparison to control and p = 0.6800 in comparison to *Gat^RNAi*, RF). **(O)** Quantification of lymph gland area in *domeMeso>GFP/+* (control, RF, n = 25), *domeMeso>GFP/Gat^RNAi* (RF, n = 17, p < 0.0001), and *domeMeso>GFP/Gat^RNAi* (Ser, n = 15, p = 0.0127 in comparison to control and p = 0.0158 in comparison to *Gat^RNAi*, RF).

Pyruvate is a 3 carbon acid and contribute its two carbons to the TCA cycle through the pyruvate dehydrogenase (PDH) dependent generation of acetyl-CoA, which combines with oxaloacetate (OAA) to form citrate and other TCA metabolites (Fig 4A). Pyruvate can additionally fuel its full 3 carbons to generate OAA through the activity of pyruvate carboxylase (PC) [51], which lies at the centre of gluconeogenesis [52,53]. Thus, the flow of labels between the two routes would indicate how lymph gland cells utilize pyruvate. The m + 2 label in OAA is derived from PDH, while m + 3 label incorporation is mediated by PC. As various steps of TCA cycle are reversible and also feed cyclically, 3 carbon label (m + 3) contributed from PC derived OAA conversion and m + 2 label contributed from PDH derived acetyl-CoA lead to higher $^{13}$C incorporation as the cycle continues and is indicative about the overall TCA cycle activity (Fig 4A).

In control lymph glands, 15% total labelling was seen in OAA. Of this, approximately 10% was m + 3 $^{13}$C labelled, which is PC-derived and 5% was m + 2, which is PDH-derived (Fig 4B). This showed PC as a favoured route for pyruvate utilization over PDH in control conditions. When assessed for TCA cycle metabolites, we observed m + 2 derived $^{13}$C-pyruvate label in citrate, αKG and malate (Fig 4C-E). Moreover, we also detected higher order label of $^{13}$C in these metabolites (Fig 4C-E) and the dynamics demonstrated pyruvate's contribution to fueling the TCA cycle. The data revealed a functional but regulated TCA cycle which aligned with our genetic findings that showcased GABA dependent regulation of PDH activity [25].

We conducted the same analysis in *domeMeso>Gat^RNAi* lymph glands where the U$^{13}$C-pyruvate label incorporation analysis revealed a rather unexpected reduction in PC derived m + 3 OAA label (Fig 4B and 4B'). PDH derived m + 2 OAA label, however, remained comparable to control lymph glands (Fig 4B). Importantly, we observed an increase in overall label incorporation in TCA metabolites with higher m + 4 isotopic carbons detected for citrate and αKG (Fig 4C-D'). Pyruvate flux towards malate however remained unchanged in *Gat^RNAi* conditions (Fig 4E-E').

The significant reduction in labelled OAA, but the concomitant increase in higher label incorporation in TCA metabolites in *domeMeso>Gat^RNAi* lymph glands implied an enhanced flux of fueling intermediates into the TCA cycle. This metabolic data corroborated with our genetic findings [25] and reinstated the importance of GABA catabolism in regulating overall TCA cycle activity. Interestingly, in the control lymph glands, our data also revealed an unexpected preference for PC as opposed to PDH for OAA production. Overall, these differences in isotopic labelling observed, while present, appear modest in magnitude. The intrinsic variability and detection sensitivity from these limiting population of lymph gland blood cells

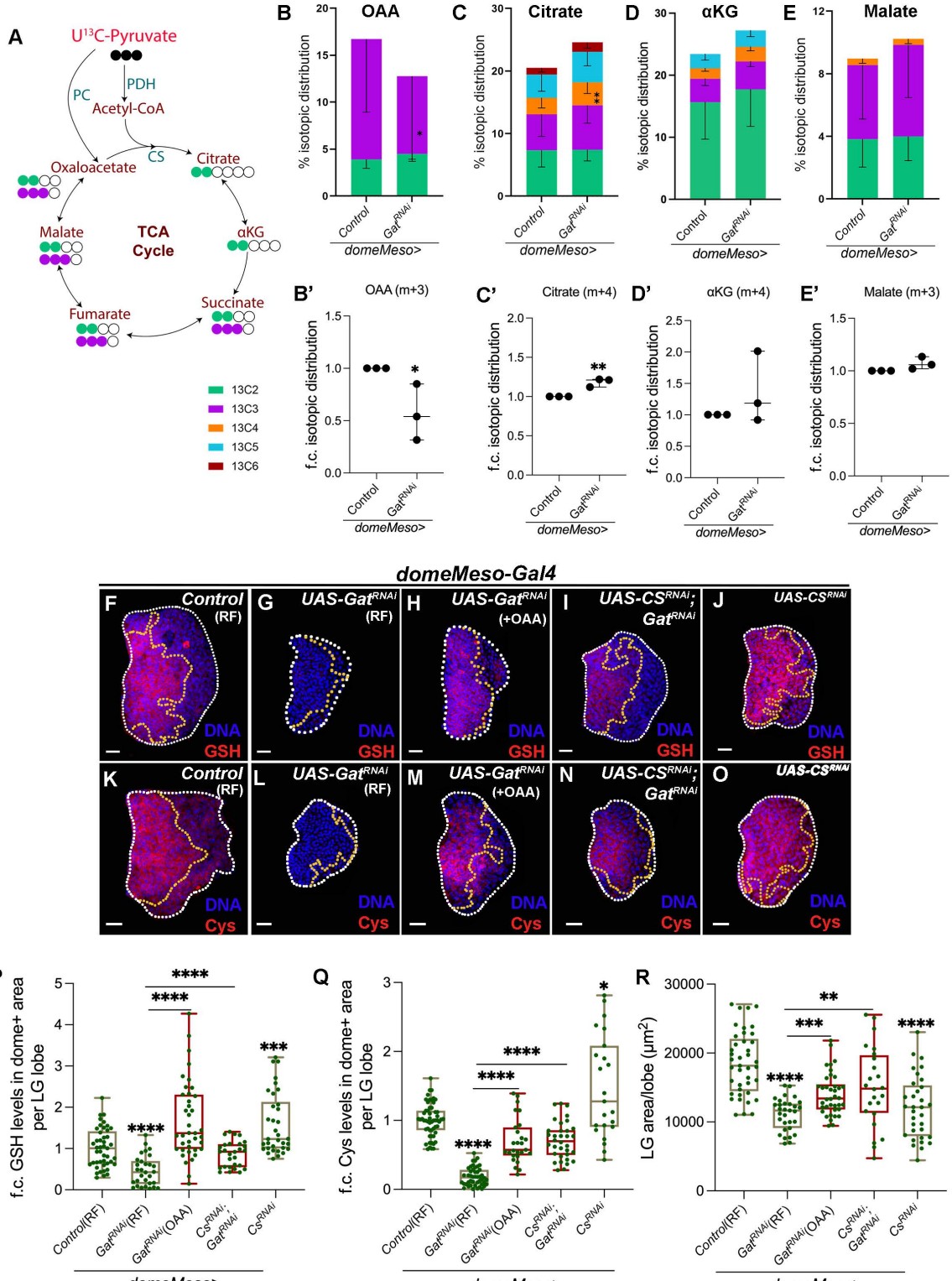

**Fig 4. GABA catabolism regulates pyruvate metabolism to maintain blood-progenitor ROS homeostasis.** Data is presented as mean with SD and stacked bar plots for **B-E** and as median plots (*p<0.05; **p<0.01; ***p<0.001; ****p<0.0001 and ns is non-significant) for **B'-E'** and **P-R**. t-test is applied for **B'-E'** and Mann-Whitney test is applied for **P-R**. In **B'-E',** 'n' is sample size and 'N' is number of experimental repeats and shown by black

dot. In **P-R**, 'n' is total number of lymph gland lobes analysed and is represented by a green dot. Scale bar: 20μm. DNA is stained with DAPI in blue. Comparisons for significance are done with control and with respective genetic conditions for rescue combinations (red bars), which are indicated by horizontal lines drawn above the box plots. White border demarcates the lymph gland lobe and yellow border marks the dome positive area towards the left side. The respective lymph gland images with dome⁺ (green) are shown in S4 Fig. **(A)** Schematic representation of the U$^{13}$C-pyruvate (black) isotopic labelling. U$_{13}$C-pyruvate contributes two 13C carbons (green) to TCA cycle metabolites via PDH and three carbons (purple) via PC mediated metabolism to OAA, which further cycles and contributes to TCA cycle metabolites. U$^{13}$C-universal 13C, all three carbons are $^{13}$C, αKG-α-ketoglutarate, CS-citrate synthase, OAA-oxaloacetate, PC-pyruvate carboxylase, PDH-pyruvate dehydrogenase, TCA cycle-tricarboxylic acid cycle. **(B-E')** Percentage (%) isotopic distribution of U$^{13}$C-pyruvate into TCA cycle metabolites, where isotopic distribution is depicted in different colours and m+2 is green, m+3 is purple, m+4 is orange, m+5 is blue and m+6 is dark red. **(B)** Control (*domeMeso>GFP/+*, n=8) and *domeMeso>GFP/Gat$^{RNAi}$* (n=8) lymph glands showing percentage label incorporation for m+2 and m+3 in OAA, **(B')** lymph glands expressing *Gat$^{RNAi}$* in the progenitor cells (*domeMeso>GFP/Gat$^{R-NAi}$*, N=3, n=8, p=0.0498) show a significant reduction for m+3 percentage label incorporation in OAA as compared to control (*domeMeso>GFP/+*, N=3, n=8), **(C)** control (*domeMeso>GFP/+*, n=8) and *domeMeso>GFP/Gat$^{RNAi}$* (n=9) lymph glands showing percentage label incorporation for m+2 to m+6 (higher isotopes) in citrate, **(C')** lymph glands expressing *Gat$^{RNAi}$* in the progenitor cells (*domeMeso>GFP/Gat$^{RNAi}$*, N=3, n=9, p=0.0042) show a significant increase for m+4 percentage label incorporation in citrate as compared to control (*domeMeso>GFP/+*, N=3, n=8), **(D)** control (*dome-Meso>GFP/+*, n=8) and *domeMeso>GFP/Gat$^{RNAi}$* (n=9) lymph glands showing percentage label incorporation for m+2 to m+5 (higher isotopes) in αKG, **(D')** lymph glands expressing *Gat$^{RNAi}$* in the progenitor cells (*domeMeso>GFP/Gat$^{RNAi}$*, N=3, n=9, p=0.3197) show an increasing trend for m+4 percentage label incorporation in αKG as compared to control (*domeMeso>GFP/+*, N=3, n=8) and **(E)** control (*domeMeso>GFP/+*, n=8) and *domeMeso>GFP/Gat$^{RNAi}$* (n=9) lymph glands showing percentage label incorporation for m+2 to m+4 (higher isotopes) in malate, **(E')** lymph glands expressing *Gat$^{RNAi}$* in the progenitor cells (*domeMeso>GFP/Gat$^{RNAi}$*, N=3, n=9, p=0.1051) does not show any significant change for m+3 percentage label incorporation in malate as compared to control (*domeMeso>GFP/+*, N=3, n=8). **(F-J)** Representative images showing GSH (red) levels in lymph gland progenitor cells (area marked within the yellow dotted line) from different genetic backgrounds. **(F)** Control (RF, *domeMeso-Gal4,UAS-GFP/+*) lymph gland showing GSH levels in progenitor-cells of the lymph gland. Expressing **(G)** *Gat$^{RNAi}$* (RF, *domeMeso-Gal4,UAS-GFP;UAS-Gat$^{RNAi}$*) in the progenitor cells leads to reduction in GSH levels as compared to control **(F)**, supplementing this genetic conditions with **(H)** oxaloacetate (OAA) restores GSH levels in *Gat$^{RNAi}$* (OAA, *domeMeso-Gal4,UAS-GFP;UAS-Gat$^{RNAi}$*) and **(I)** progenitor specific expression of *Cs$^{RNAi}$* in *Gat$^{RNAi}$* animals (*domeMeso-Gal4,UAS-GFP; UAS-Cs$^{RNAi}$;UAS-Gat$^{RNAi}$*) leads to a recovery of blood-progenitor GSH levels, compare with **(G)** GSH in *Gat$^{RNAi}$*. **(H)** Progenitor-specific expression of *Cs$^{RNAi}$* (*domeMeso-Gal4,UAS-GFP; UAS-Cs$^{RNAi}$*) leads to increase in GSH levels as compared to control **(F)**. For quantifications, refer to **P**. **(K-O)** Representative images showing cysteine (Cys, red) levels in lymph gland progenitor cells (area marked within the yellow dotted line) from different genetic backgrounds. **(K)** Control (RF, *domeMeso-Gal4,UAS-GFP/+*) lymph gland showing Cys levels in progenitor-cells of the lymph gland. Expressing **(L)** *Gat$^{RNAi}$* (RF, *domeMeso-Gal4,UAS-GFP;UAS-Gat$^{RNAi}$*) in the progenitor cells leads to reduction in Cys levels as compared to control **(K)**, supplementing this genetic conditions with **(M)** oxaloacetate (OAA) restores Cys levels in *Gat$^{RNAi}$* (OAA, *domeMeso-Gal4,UAS-GFP;UAS-Gat$^{RNAi}$*) and **(N)** progenitor specific expression of *Cs$^{RNAi}$* in *Gat$^{RNAi}$* animals (*domeMeso-Gal4,UAS-GFP; UAS-Cs$^{RNAi}$;UAS-Gat$^{RNAi}$*) leads to a recovery of blood-progenitor Cys levels, compare with **(L)** GSH in *Gat$^{RNAi}$*. **(O)** Progenitor-specific expression of *Cs$^{RNAi}$* (*domeMeso-Gal4,UAS-GFP; UAS-Cs$^{RNAi}$*) leads to increase in Cys levels as compared to control **(K)**. For quantifications, refer to **Q**. **(P)** Quantification of blood-progenitor GSH levels (fold change, f.c.) in *domeMeso>GFP/+* (control, RF, n=45), *domeMeso>GFP/Gat$^{RNAi}$* (RF, n=30, p<0.0001), *domeMeso>GFP/Gat$^{RNAi}$* (OAA, n=39, p<0.0001 in comparison to *Gat$^{RNAi}$*, RF), *domeMeso>GFP/Cs$^{RNAi}$;Gat$^{RNAi}$* (n=28, p<0.0001 in comparison to *Gat$^{RNAi}$*) and *domeMeso>GFP/Cs$^{RNAi}$* (n=34, p=0.0003). **(Q)** Quantification of blood-progenitor cysteine (Cys) levels (fold change, f.c.) in *domeMeso>GFP/+* (control, RF, n=55), *domeMeso>GFP/Gat$^{RNAi}$* (RF, n=46, p<0.0001), *domeMeso>GFP/Gat$^{RNAi}$* (OAA, n=27, p<0.0001 in comparison to *Gat$^{RNAi}$*, RF), *domeMeso>GFP/Cs$^{RNAi}$;Gat$^{RNAi}$* (n=34, p<0.0001 in comparison to *Gat$^{R-NAi}$*) and *domeMeso>GFP/Cs$^{RNAi}$* (n=23, p=0.0275). **(R)** Quantification of lymph gland area in *domeMeso>GFP/+* (control, RF, n=41), *domeMeso>GFP/Gat$^{RNAi}$* (RF, n=31, p<0.0001), *domeMeso>GFP/Gat$^{RNAi}$* (OAA, n=38, p=0.0003 in comparison to *Gat$^{RNAi}$*, RF), *domeMeso>GFP/Cs$^{RNAi}$;Gat$^{RNAi}$* (n=24, p=0.0037 in comparison to *Gat$^{RNAi}$*) and *domeMeso>GFP/Cs$^{RNAi}$* (n=33 p<0.0001).

may contribute to the relatively high standard deviations leading to minor differences. Nevertheless, these differences observed were consistent across replicates and formed the basis of our working model.

We speculate that in homeostasis, the PC derived m+3 labelled OAA supports *de novo* synthesis of serine through gluconeogenic routes and capacitates progenitor cells with redox regulating potential. In the absence of GABA catabolism, the shift in pyruvate flux towards PDH-mediated entry into the TCA cycle, results in elevated citrate synthesis and reduced OAA availability. This metabolic re-routing may lead to impaired serine biosynthesis and downstream GSH generation, and compromises progenitor redox homeostasis.

To test whether limited OAA availability contributed to the reduction in progenitor GSH synthesis observed in the *domeMeso>Gat$^{RNAi}$* condition, we supplemented these animals with exogenous OAA and assessed GSH levels. Remarkably, OAA supplementation restored both progenitor GSH (Figs 4F-H, 4P and S4J-L) and cysteine levels (Figs 4K-M, 4Q and S4O-Q) in *Gat$^{RNAi}$* expressing lymph glands, supporting a critical role for OAA in driving GSH biosynthesis via GABA metabolism. To further probe this mechanism, we asked whether the heightened routing of OAA into the TCA cycle reflected by elevated citrate levels in *Gat$^{RNAi}$* glands was limiting OAA availability for serine and GSH biosynthesis. To

reduce OAA utilization within the TCA cycle, we genetically downregulated *citrate synthase* (*Cs*) in the *Gat^RNAi* background. Strikingly, this combination restored both GSH (Figs 4I, 4P and S4M) and cysteine levels (Figs 4N, 4Q and S4R) in blood progenitors. Notably, *Cs* knockdown alone (*domeMeso>Cs^RNAi*) was also sufficient to enhance progenitor GSH (Figs 4J, 4P and S4N) and Cys levels (Figs 4O, 4Q and S4S). These genetic results closely phenocopied the effects of OAA supplementation in *Gat^RNAi* animals. Furthermore, both OAA supplementation and *Cs* downregulation in the *Gat^RNAi* background also rescued the lymph gland growth defects associated with GABA loss (Fig 4R) conditions.

Together with mass spectrometry-based metabolic flux analysis, these findings support a model wherein excessive routing of OAA toward citrate generation, rather than its metabolic diversion toward serine and GSH biosynthesis, underlies the reduced GSH levels in the *Gat^RNAi* condition. Thus, GABA catabolism plays a pivotal role in regulating TCA cycle activity and enabling the availability of OAA for redox-regulatory biosynthetic processes in progenitor cells.

## Oxaloacetate flux into gluconeogenic arm and serine synthesis regulate GSH homeostasis

Downstream of oxaloacetate, its driving of gluconeogenesis can occur via phosphoenolpyruvate carboxykinase (PEPCK) enzyme, either PEPCK1 or PEPCK2, to fuel serine and downstream GSH synthesis [54,55]. Specifically, OAA is converted into phosphoenol pyruvate (PEP) which subsequently gives rise to 3-phosphoglycerate (3-PG), a glycolytic/gluconeogenic intermediate. 3-PG undergoes a three-step conversion into serine, the first and rate-limiting step being catalyzed by phosphoglycerate dehydrogenase (PHGDH, Fig 5A) [49,56].

To test the functional implications of this pathway, we independently down-regulated these key rate limiting enzymes, *Pepck1*, *Pepck2* and *Phgdh* in the blood progenitor cells and assessed their impact on GSH and cysteine levels, ROS status and lymph gland size.

While *Pepck1* knockdown had no significant impact on GSH levels or lymph gland size (Fig S5A-B' and 5P-Q), *Pepck2* knockdown (*domeMeso>Pepck2^RNAi*) caused a pronounced reduction in progenitor GSH levels (Figs 5B-C, 5L and S5C-D), elevation in ROS levels (Figs 5D-E, 5M and S5E-F) and lymph gland size reduction (Fig 5N). This suggests a specific role of mitochondrial PEPCK2 mediated OAA conversion in GSH biosynthesis. Correspondingly, *Pepck2* downregulation also reduced progenitor cysteine levels (S5M-N' and S5R Fig).

Knockdown of *Phgdh* (*domeMeso>Phgdh^RNAi*) similarly led to reduced progenitor GSH levels (Figs 5F, 5L and S5G), increased ROS (Figs 5G, 5M and S5H), and smaller lymph glands (Fig 5N). Interestingly, *Phgdh* knockdown did not reduce cysteine levels and it remained comparable to controls (S5O-O' and S5R Fig). This discrepancy may arise due to the nature of anti-cysteine antibody used, which detects cysteinylated proteins as opposed to free cysteine. Alternatively, serine also contributes to glycine synthesis (via Serine Hydroxymethyltransferase, SHMT) and glycine is also a component of GSH, so reduced serine might impair GSH levels through glycine deficiency.

To further investigate this pathway, we supplemented *Pepck2^RNAi* and *Phgdh^RNAi* expressing lymph glands with serine. This showed recovery in progenitor GSH (S6A-E' and S6M Fig), cysteine levels (S6F-J' and S6N Fig), and also rescued lymph gland size defect (S6O Fig). Next, in *Pepck2^RNAi* and *Phgdh^RNAi* backgrounds, N-acetylcysteine (NAC) supplementation was undertaken, which was also sufficient to restore progenitor GSH (Figs 5H-I, 5L and S5I-J), ROS (Figs 5J-K, 5M and S5K-L) and lymph gland size (Fig 5N), thereby reinforcing the importance of the OAA-serine-cysteine axis in redox regulation and lymph gland growth dynamics.

Additionally, the *Phgdh^RNAi* animals were supplemented with glycine, as they showed no change in cysteine. This also led to significant recovery of GSH, further elevation in cysteine levels. This suggested that serine-derived cysteine and glycine contributed critically to GSH biosynthesis. Notably, unlike serine (S6E and S6O Fig) and cysteine (Fig 5I and 5N) supplementation, glycine supplementation failed to produce a comparable or consistent recovery in lymph gland size (S6K and S6O Fig). This suggests that while glycine supports GSH synthesis, it has a limited capacity to restore progenitor proliferation or growth on its own, underscoring the broader metabolic roles of serine and cysteine in progenitor development.

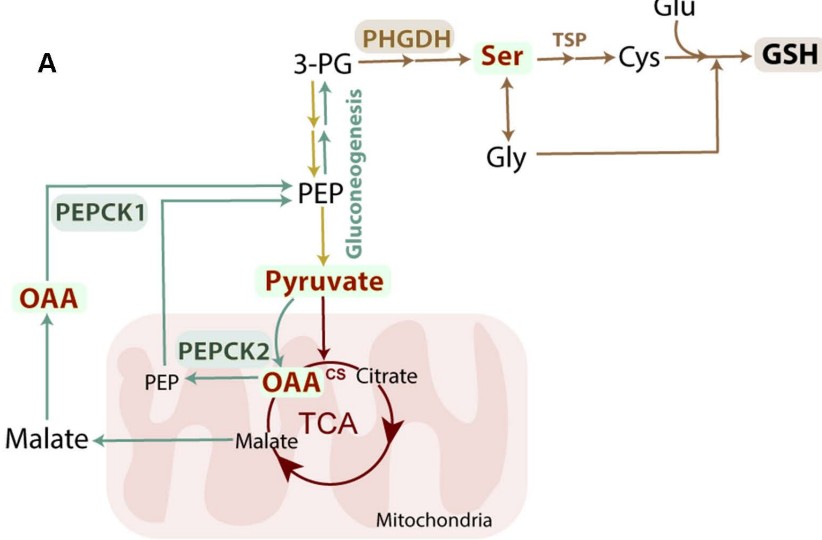

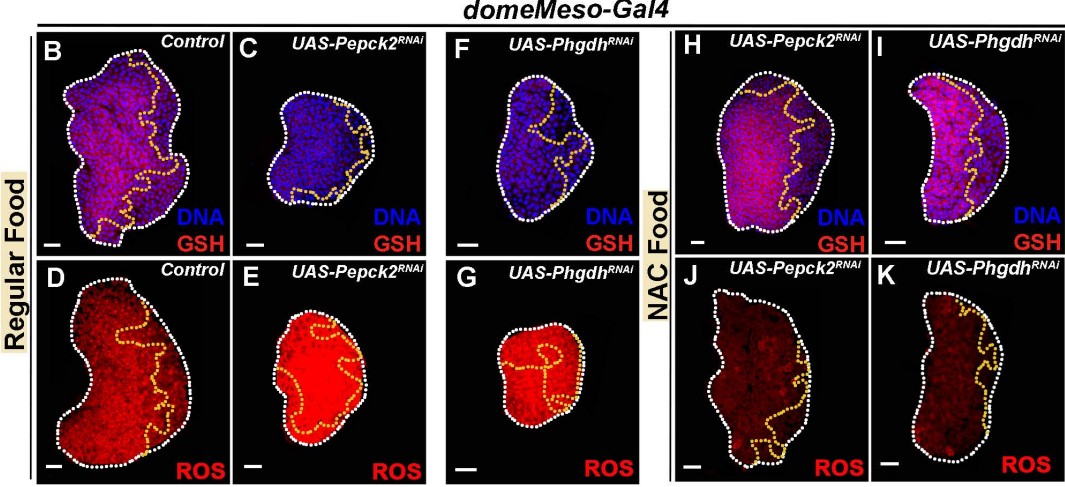

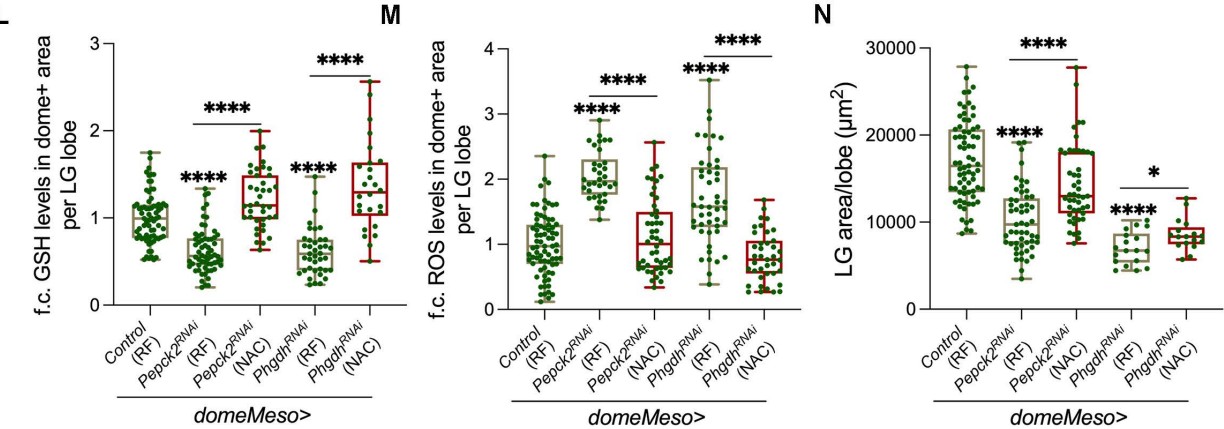

**Fig 5. Oxaloacetate (OAA) flux into gluconeogenic arm and serine synthesis regulate GSH homeostasis.** RF is regular food and NAC is N-acetylcysteine supplemented food. Data is presented as median plots (**p < 0.01 and ****p < 0.0001). Mann-Whitney test is applied for **L-N**. In **L-N**, 'n' is total number of lymph gland lobes analysed and is represented by a green dot. Scale bar: 20μm. DNA is stained with DAPI in blue. Comparisons

for significance are done with control and with respective genetic conditions for rescue combinations (red bars), which are indicated by horizontal lines drawn above the box plots. White border demarcates the lymph gland lobe and yellow border marks the dome positive area towards the left side. The respective lymph gland images with dome⁺ (green) are shown in S5 Fig. **(A)** Schematic representation of OAA flux into gluconeogenesis leading to serine and GSH generation. OAA formed in the cells from TCA cycle and PC facilitates gluconeogenesis by PEPCK1 (cytoplasmic) and PEPCK2 (mitochondrial), which produce PEP. PEP converts into 3-PG, a gluconeogenic intermediate, that fuels Ser synthesis by a three step process, the first step catalysed by enzyme PHGDH. Ser leads to Cys formation through TSP and also converts into Gly by one-carbon metabolism. Cys and Glu conjugate to form γ-glutamyl-cysteine, and further addition of Gly leads to GSH formation. 3-PG-3-phosphoglycerate, Cys-cysteine, Glu-glutamate, Gly-glycine, GSH-glutathione, OAA-oxaloacetate, PC-pyruvate carboxylase, PEP-phosphoenol pyruvate, PEPCK1-phosphoenolpyruvate carboxykinase 1, PEPCK2- phosphoenolpyruvate carboxykinase 2, PHGDH-phosphoglycerate dehydrogenase, TCA cycle-tricarboxylic acid cycle, TSP-transsulfuration pathway. **(B-G)** Representative images showing glutathione (GSH, red) and ROS (red) levels in lymph gland progenitor cells (area marked within the yellow dotted line) across different genetic backgrounds. **(B)** control (*domeMeso-Gal4,UAS-GFP*/+) lymph gland showing GSH levels, expressing **(C)** *Pepck2^RNAi^* (*domeMeso-Gal4,UAS-GFP;UAS-Pepck2^RNAi^*) in the progenitor cells leads to reduction in blood-progenitor GSH levels. **(D)** control (*domeMeso-Gal4,UAS-GFP*/+) lymph gland showing ROS levels, expressing **(E)** *Pepck2^RNAi^* (*domeMeso-Gal4,UAS-GFP;UAS-Pepck2^RNAi^*) in the progenitor cells leads to elevation of blood-progenitor ROS levels. Similarly, expressing **(F-G)** *Phgdh^RNAi^* (*domeMeso-Gal4,UAS-GFP;UAS-Phgdh^RNAi^*) leads to reduction in blood-progenitor **(F)** GSH and elevation of **(G)** ROS levels. Compare to control **(B)** GSH and **(D)** ROS. For quantifications, refer to **L** and **M**. **(H-K)** NAC supplementation to **(H, J)** *Pepck2^RNAi^* (NAC, *domeMeso-Gal4,UAS-GFP;UAS-Pepck2^RNAi^*) and **(I, K)** *Phgdh^RNAi^* (NAC, *domeMeso-Gal4,UAS-GFP;UAS-Phgdh^RNAi^*) leads to recovery of blood progenitor **(H, I)** GSH and **(J, K)** ROS levels. For comparison refer to **(C, D)** GSH and **(F, G)** ROS in *Pepck2^RNAi^* (RF, *domeMeso-Gal4,UAS-GFP;UAS-Pepck2^RNAi^*) and *Phgdh^RNAi^* (RF, *domeMeso-Gal4,UAS-GFP;UAS-Phgdh^RNAi^*) respectively. For quantifications, refer to **L** and **M**. **(L)** Quantification of blood-progenitor GSH levels (fold change, f.c.) in *domeMeso>GFP*/+ (control, RF, n=72), *domeMeso>GFP/Pepck2^RNAi^* (RF, n=62, p<0.0001), *domeMeso>GFP/Pepck2^RNAi^* (NAC, n=39, p<0.0001 in comparison to *Pepck2^RNAi^*, RF), *domeMeso>GFP/Phgdh^RNAi^* (RF, n=39, p<0.0001) and *domeMeso>GFP/Phgdh^RNAi^* (NAC, n=26, p<0.0001 in comparison to *Phgdh^RNAi^*, RF). **(M)** Quantification of blood-progenitor ROS levels (fold change, f.c.) in *domeMeso>GFP*/+ (control, RF, n=75), *domeMeso>GFP/Pepck2^RNAi^* (RF, n=34, p<0.0001), *domeMeso>GFP/Pepck2^RNAi^* (NAC, n=48, p<0.0001 in comparison to *Pepck2^RNAi^*, RF), *domeMeso>GFP/Phgdh^RNAi^* (RF, n=47, p<0.0001) and *domeMeso>GFP/Phgdh^RNAi^* (NAC, n=39, p<0.0001 in comparison to *Phgdh^RNAi^*, RF). **(N)** Quantification of lymph gland area in *domeMeso>GFP*/+ (control, RF, n=71), *domeMeso>GFP/Pepck2^RNAi^* (RF, n=53, p<0.0001), *domeMeso>GFP/Pepck2^RNAi^* (NAC, n=47, p<0.0001 in comparison to *Pepck2^RNAi^*, RF), *domeMeso>GFP/Phgdh^RNAi^* (RF, n=18, p<0.0001) an d*omeMeso>GFP/Phgdh^RNAi^* (NAC, n=17, p=0.0269 in comparison to *Phgdh^RNAi^*, RF).

Altogether, our findings reinforced the importance of the OAA-serine-cysteine axis in redox regulation. OAA serves a critical role beyond the TCA cycle, feeding into gluconeogenesis via PEPCK2 to generate serine and maintain GSH homeostasis in blood progenitors. Importantly, these effects are largely specific to the medullary zone, as rescue effects with GABA-related metabolites (succinate, NAC, OAA, serine, glycine) were most evident there. Thus, we propose a model where GABA catabolism regulates TCA cycle flux and ensures OAA availability for serine/GSH synthesis to maintain blood progenitor ROS homeostasis and proper lymph gland development.

## Discussion

Blood progenitor cells in the *Drosophila* lymph gland depend on reactive oxygen species (ROS) to preserve their differentiation potential. However, to sustain an undifferentiated state and avoid precocious differentiation, these cells must also tightly regulate their redox homeostasis [5]. The metabolic processes that coordinate ROS production with antioxidant capacity form the central focus of this study.

### Proposed model

Including mass spectrometry data and genetic interaction studies, presented in the current study, we propose the following model (Fig 6A): In homeostasis, lymph gland progenitor cells metabolize pyruvate into oxaloacetate (OAA) via pyruvate carboxylase (PC) and through pyruvate dehydrogenase (PDH), where the former is favoured over the later. This restriction on PDH serves to limit excessive flux into the TCA cycle and preserves OAA for other biosynthetic functions. Specifically, OAA in the lymph gland is routed through the gluconeogenic pathway via mitochondrial PEPCK2 enzyme, facilitating the production of serine, a precursor for cysteine and glycine, both essential components of glutathione (GSH) synthesis.

Thus, by modulating pyruvate metabolism at the PC/PDH junction, progenitors balance TCA cycle activity with antioxidant synthesis, enabling fine control of their redox state. Importantly, GABA catabolism plays a central role in this regulation. The GABA shunt limits PDH activity and supports PC-mediated flux into OAA, thereby promoting serine biosynthesis

A

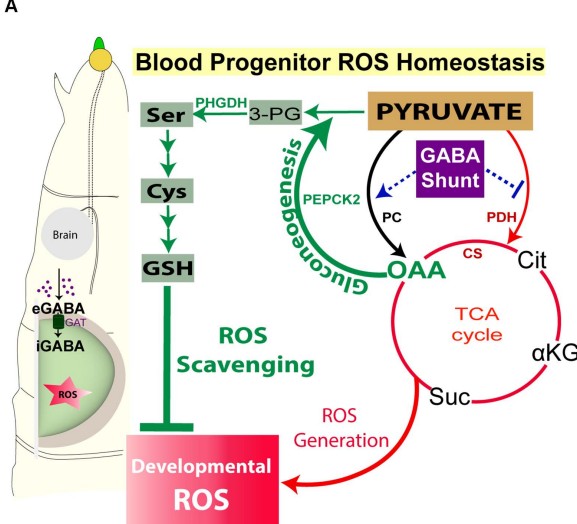

**Fig 6. Control of *de novo* GSH synthesis in lymph gland blood progenitor cells by the GABA shunt pathway. (A)** Pyruvate is metabolised into OAA by PC and PDH. GABA controls PDH activity and restricts TCA cycle flux. This state is necessary to support GSH production in the progenitor cells. Pyruvate's availability for gluconeogenesis via OAA and its metabolism to 3-PG mediated by PEPCK2 is the key step to allow *de novo* synthesis of serine. Serine is a key intermediate for promoting GSH production in these cells. By restraining PDH activity and positively influencing serine production, GABA catabolism enables a state in blood progenitor cells, whereby they are able to generate ROS, as well as capacitate them with ROS scavenging potential to counter any redox stress. This metabolic program maintains homeostatic ROS levels in the lymph gland, which sensitizes progenitors to differentiation cues [5], but also limits excessive ROS production to support lymph gland growth [25]. αKG-α-ketoglutarate, Cit-citrate, CS-citrate synthase, Cys-cysteine, eGABA-extracellular GABA, GABA-γ-aminobutyric acid, GAT-GABA Transporter, GSH-glutathione, iGABA-intracellular GABA, OAA-oxaloacetate, PC-pyruvate carboxylase, PDH-pyruvate dehydrogenase, PHGDH-phosphoglycerate dehydrogenase, ROS-reactive oxygen species, Ser-serine, Suc-succinate, TCA cycle-tricarboxylic acid cycle.

and sustaining GSH production. In the absence of GABA metabolism, this balance is disrupted; PDH activity is upregulated, leading to increased routing of pyruvate into the TCA cycle, excessive ROS production, and reduced availability of OAA for serine and GSH biosynthesis. This metabolic shift results in redox imbalance and impaired lymph gland development. Restoring redox balance; either by inhibiting PDH, limiting OAA's entry into the TCA cycle, or supplementing with OAA, serine, or GSH precursors, rescues progenitor function. Altogether, our findings uncover a critical nexus in blood progenitor cells where ROS generation and scavenging are metabolically intertwined. GABA metabolism acts as a pivotal regulator of this balance, orchestrating pyruvate flux to maintain ROS homeostasis and progenitor cell fate.

## TCA Cycle, GABA shunt and redox regulation

The TCA cycle is a critical metabolic hub that modulates ROS levels in progenitor cells. Pyruvate's entry into the TCA cycle through PDH, a rate-limiting enzyme that converts pyruvate to acetyl-CoA, is the primary route for ROS production in these cells [25]. Blood progenitor cells tightly regulate PDH activity to prevent excessive ROS accumulation. Previous studies have demonstrated that GABA catabolism plays a key role in this regulation, with succinate derived from GABA metabolism modulating PDH activity and maintaining ROS homeostasis.

Our metabolic flux measurements reveal PC as an additional route for pyruvate metabolism, which is impaired in the absence of GABA. GABA catabolism fine-tunes TCA cycling by regulating PDH activity and limiting intermediate utilization, such as OAA, within the TCA cycle. OAA's availability is necessary for gluconeogenic programs, including serine biosynthesis. In GABA catabolic loss-of-function condition, increased PDH activity prioritizes intermediate utilization, leading to a lack of OAA for other metabolic processes, creating a bottleneck that compromises antioxidant synthesis. Accelerated TCA cycling depletes intermediates needed for serine and GSH biosynthesis, tipping the redox balance toward oxidative stress.

Dysregulation of TCA cycle flux, such as succinate accumulation, has been linked to impaired redox homeostasis [57]. The GABA shunt, in this regard, is often seen as a bypass of the conventional TCA cycle, where it plays a significant role in TCA regulation. GABA is metabolized into succinate through the sequential action of GABA transaminase and succinate semialdehyde dehydrogenase (SSADH). Succinate enters the TCA cycle directly, bypassing the key steps like αKDH and PDH that contribute to ROS generation. Thus, by providing an alternate source of TCA cycle intermediates, GABA shunt fine-tunes the rate of TCA cycling and mitigates excessive ROS production. However, in the lymph gland blood progenitor cells the catabolism of GABA into succinate functions in a completely different way. We find GABA-derived succinate to be closely associated with its role in the post-translational regulation of protein function. Either through regulation of Hph and thereby stabilizing Hif-1α [26], or by controlling PDK activity and PDH phosphorylation [25], GABA shunt operates to control key enzymatic steps that are rate limiting to either glycolysis or TCA cycle activity. The influence of shunt in the production of OAA by regulating PC activity, is yet another revelation in this study and positions GABA shunt pathway at the nexus of ensuring a balance between energy production and biosynthetic needs within the progenitor cells.

### Blood progenitor reliance on Serine/GSH *de novo* biosynthesis via gluconeogenesis: a means to enable adaptability to stress conditions

Serine, a precursor for GSH synthesis, is produced from glycolytic intermediates and is directly influenced by TCA cycle activity. TCA derived α-ketoglutarate and malate promotes serine synthesis by activating key enzymes in the pathway, such as phosphoglycerate dehydrogenase (PHGDH) [58]. PHGDH activity is sensitive to the NAD/NADH ratio, which TCA cycle metabolites maintain through pyruvate-to-lactate conversion and the malate-aspartate shuttle [59]. In macrophages, dysregulation of *de novo* serine synthesis impairs GSH synthesis and ROS homeostasis leading to loss of inflammatory response in them [60].

*De novo* serine synthesis in blood progenitor cells and its contribution to GSH synthesis by serving as a carbon donor for cysteine synthesis is very striking. Moreover, the influence of GABA sustaining this state is even more impressive. We know that in the blood progenitor cells, GABA catabolism through stabilizing Hif-1α and LDH activity, promotes their glycolytic state, which is necessary to support their immune competency [26,61]. Thus GABA functions in immune modulation and maintains a pro-inflammatory state in progenitor cells. While the sustenance of a glycolytic state may allow the NAD/NADH ratios to be maintained to support serine synthesis, the implications of maintaining a *de novo* route to GSH generation may be a part of the larger immune competency program that GABA catabolism supports. The purpose of utilizing glycolytic intermediates to make serine and GSH, may be an option to counter oxidative stresses during infection and avoid any progenitor cell death. Serine metabolism is also closely linked to one-carbon metabolism where it integrates nutritional status and generates diverse outputs, including methylation substrates [55]. For example, glucose-dependent serine biosynthesis supports T-cell proliferation [62], while TCA dependent serine synthesis via PEPCK2 is central to metabolic adaptation in tumor cells [54]. Intriguingly, key enzymes for serine biosynthesis and PEPCK2 exhibit similar expression patterns during skeletal muscle cell development [63] and moreover, PEPCK2 is shown to promote oxaloacetate derived *de novo* serine synthesis [54]. This clearly implies that serine biosynthesis depends critically on PEPCK dependent diversion of TCA cycle intermediates [64]. Thus, keeping this route to generate serine and GSH via gluconeogenic entry of OAA may also allow the progenitor cells to maintain a metabolic state that integrates inputs from multiple nutritional routes, ensuring adaptability and resilience under oxidative environmental stressors.

### Implications for hematopoietic systems and conclusions

These blood progenitor cells of the lymph gland, like the common myeloid precursors are highly dependent on stress signals for their development. They develop in niches with high ROS [5,21], high calcium [13,65] and respond promptly to any changes in their levels. Thus, these cells have optimised the use of stress signals to program their development. In this regard, our study alludes to the underlying progenitor-specific metabolic program that ensures their adaptability and

resilience to such developmental stressors. This notion is supported by multiple pieces of published evidences that show how impairing GABA function in progenitor cells, affects their ability to respond to environmental changes, that include sensory stimulation, infection and also changes to their local niche environment, like alteration in calcium [13] or ROS homeostasis [25]. The control of GABA shunt in optimising pyruvate metabolism, to ensure a multifaceted development of the blood-progenitor cells is indeed the most prominent phenomenon that emerges from our past and current study.

Mammalian hematopoietic progenitor cells exhibit a similar reliance on serine biosynthesis [66] and GABA/GABA-receptor signaling [34]. ROS signaling plays a crucial role in hematopoietic stem cell (HSC) proliferation, with NAC treatments restoring HSC quiescence and proliferation under oxidative stress [35]. These commonalities lead us to speculate similar modalities for metabolic regulation in mammalian hematopoietic stem cell niches as well. The optimal application of this metabolic framework to safeguard differentiation potential under oxidative stress is striking and could be a conserved phenomenon that however remains to be investigated. Overall, the intricate interplay between metabolic and redox regulation during development and the integral role of GABA shunt in keeping the checks in place positions GABA in an entirely new developmental context, as a redox modulator in stem/progenitor cells.

## Materials and methods

### *Drosophila* husbandry, stocks, and genetics

The following *Drosophila melanogaster* stocks were used in this study: $w^{1118}$ (wild type, *wt*) and *domeMeso-Gal4,UAS-GFP* (U. Banerjee). The RNA*i* stocks were either obtained from VDRC (Vienna) or BDSC (Bloomington) *Drosophila* stock centres. The lines used in this study are: $Gat^{RNAi}$ (BDSC 29422), $Ssadh^{RNAi}$ (VDRC 106637), $Pdha^{RNAi}$ (BDSC 55345) $Pdk^{RNAi}$ (BDSC 28635), $Cs^{RNAi}$ (BDSC 60900, [67]), $Pepck1^{RNAi}$ (BDSC 65087, [68]), $Pepck2^{RNAi}$ (VDRC 13929, [68,69]) and $Phgdh^{RNAi}$ (BDSC 57822).

All fly stocks were reared on corn meal agar food medium with yeast supplementation at 25°C incubator unless specified. Tight collections were done for 4–6 hours to avoid over-crowding and for synchronous development of larvae. The crosses involving RNA*i* lines were maintained at 29°C to maximize the efficacy of the *Gal4/UAS$^{RNAi}$* system. Controls correspond to *Gal4* drivers crossed with $w^{1118}$. FlyBase was used throughout the study to find information on genes, pathways and available fly stocks [70].

*Gal4* efficiency for each RNA*i* line and the co-expression RNA*i* lines was validated by both antibody staining for genes downregulated in the RNA*i* lines, which showed a significant downregulation for PDH and Gat protein in the respective genetic combinations (S3 Fig), and checking the GFP intensity from the *UAS-GFP* expression driven under the control of *dome>Meso-Gal4*. GFP intensity remained comparable across different genetic combinations (S6P Fig).

### Immunostaining and immunohistochemistry

Immunohistochemistry on lymph gland tissues were performed with the following primary antibodies: rabbit-αGlutathione (GSH, 1:100, abcam #ab9443), mouse-αGlutamate (Glu, 1:100, Sigma G9282) and mouse-αCysteine (Cys, 1:20, sc-69954), mouse-αPDH (Abcam, ab110334, 1:250), Rabbit-αGat (1:5000, M. Freeman) and rabbit-αpPDH (Abcam, S293, ab177461, 1:250). The secondary antibodies Alexa Flour 488, 546 and 647 (Invitrogen) were used at 1:400 dilutions. Nuclei were visualized using DAPI (Sigma). Samples were mounted with Vectashield (Vector Laboratories).

Lymph glands dissected from wandering 3rd instar larvae were stained following the protocol of [71]. Lymph gland tissues from synchronized larvae of required developmental stage were dissected in cold PBS (1X Phosphate Buffer Saline, pH-7.2) and fixed in 4% formaldehyde for 40 min. at room temperature. Tissues were then washed thrice (15 min. each wash) in 0.3% PBT (0.3% triton X-100 in 1X PBS) for permeabilization and further blocking was done in 5% NGS (Normal Goat Serum, Jackson Laboratories), for 45 min at RT. After this, tissues were incubated in the respective primary antibodies with appropriate dilution in 5% NGS overnight at 4°C. Post primary antibody incubation, tissues were washed thrice in 0.3% PBT for 15 min. each. This was followed by incubation of tissues in respective secondary antibodies for 2–3 hrs at

RT. After secondary antibody incubation, tissues were washed in 0.3% PBT for 15 min. following a DAPI+0.3% PBT wash for 15 min. Excess DAPI was washed off by a wash of 0.3% PBT for 15 min. Tissues were mounted in Vectashield (Vector Laboratories) and then imaged utilizing confocal microscopy. For representation, one lymph gland lobe is shown in the figure panels.

For mild washing of tissues, to detect small metabolites, dissected lymph glands were fixed in 4% formaldehyde and processed for washing with 0.1% tween-20 in 1XPBS instead of 0.3% triton X-100 in 1XPBS.

### ROS (DHE) detection in lymph glands

Lymph glands dissected from the wandering 3rd instar larvae were stained for ROS levels following the protocol of [72]. The dissected lymph gland tissues were stained in 1:1000 DHE (Invitrogen, Molecular Probes, D11347) dissolved in 1X PBS for 15 min in the dark. Tissues were washed in 1X PBS twice and fixed with 4% formaldehyde for 6–8 min. at room temperature in the dark. Tissues were again quickly washed in 1X PBS twice and then mounted in Vectashield (Vector Laboratories). The lymph glands were imaged immediately.

### Image acquisition and processing

Immuno-stained and DHE stained (ROS) lymph gland tissue images were acquired using Olympus FV3000 Confocal Microscopy 40X oil-immersion objective. Microscope settings were kept constant for each sample in every experiment. The image acquisition settings were chosen to capture this difference in control lymph glands without causing saturation in majority of the pixels. This setting was thereafter kept constant for all other genotypes that were conducted in the corresponding experimental batch and were processed for analysis and quantifications. Lymph gland images were processed using ImageJ (Fiji) and Adobe Photoshop software.

### Quantification of lymph gland phenotypes

All images were quantified using ImageJ (Fiji) software and Microsoft Excel. Images were acquired as z-stacks and quantifications were done as described previously [25]. For antibody intensity and ROS levels quantifications, the area covering the dome+ region of the lymph gland lobe was marked utilizing free hand tool in ImageJ and mean fluorescence intensity was calculated. Background noise was quantified marking four random square of equal size in each image and corresponding average intensities were subtracted from the mean intensity values calculated from the respective signal. The relative fold change was calculated from the final mean fluorescence intensity values in Microsoft Excel and graph plotting, and statistical data analysis was performed using GraphPad Prism software. For all intensity quantifications, the laser settings for each individual experimental set-up were kept constant and controls were analysed in parallel to the RNAi conditions every time. For lymph gland area analysis, middle two z-stacks of the image were merged, and total lymph gland lobe area was marked using the free-hand tool of ImageJ and then analysed further for quantifications. In all experiments "n" implies the total number of lymph gland lobes analysed which were obtained from multiple independent experiments, repeated at least three times.

### Metabolite supplementation

Glutathione reduced (GSH, G4251), Succinate (SF, Sodium succinate dibasic hexahydrate, Sigma, S9637), N-Acetyl-L-cysteine (NAC, Sigma, A7250), methionine (Met, Sigma, M9625), serine (Ser, Sigma, S4500), OAA (Sigma, O4126) and glycine (Gly, Sigma, G7126) enriched diets were prepared by supplementing regular fly food with weight/volume measures of each supplement and 0.1% GSH, 3% Succinate, 0.1% NAC, 0.1% methionine, 0.1% serine, 0.2% OAA and 0.1% glycine concentrations were used. Eggs were transferred in these supplemented diets and reared until analysis of the lymph gland tissues at the wandering 3rd instar stage.

## Metabolite extraction and targeted metabolomic analysis

For metabolite extraction, five lymph glands from wandering 3$^{rd}$ instar larvae were taken per sample in 200 µl of 80% ice-cold methanol and samples were homogenized briefly. Samples were stored at -80 degrees immediately and later dried down in a Vacufuge plus speed-vac at room temperature and derivatized further with OBHA/EDC for metabolite analysis [73,74]. For derivatization, dried samples were dissolved in 50 µl of LC/MS grade water and 50 µl of 1M EDC (in Pyridine buffer, pH 5) was added. These samples were kept on a thermomixer for 10 min. at room temperature and 100 µl of 0.5M OBHA (in Pyridine buffer, pH 5) was added. The samples were incubated again for 1.5 hours on the thermomixer at 25°C, and metabolites were extracted by adding 300 µl of ethyl acetate and this step was repeated three times. Samples were dried down in a Vacufuge plus speed-vac at room temperature and stored at -80°C until run for LC/MS analysis. The metabolite extract was separated using a Waters XBridge C18 Column (2.1 mm, 100 mm, 3.5 mm) on an Agilent QQQ 6470 system coupled to a Agilent 1290 UPLC system. The autosampler and column oven were kept at 4°C and 25°C, respectively. The buffers utilised for the analysis were, buffer A: Water plus 0.1% Formic Acid and buffer B 100% acetonitrile (ACN) plus 0.1% Formic Acid. A flow rate of 0.300 ml/minute was used for the chromatographic gradient as follows: 0 min-10% B; 0.50 min: 10% B; 8 min: 100% B; 10 min: 10% B; 11 min: 10% B and at 16 min gradient was held at 10%B. MRM, positive ion mode was used for running the LC/MS and mass spectrometry detection was carried out on a QQQ Agilent 6470 system with ESI source attached to a UPLC system. For metabolite quantification, Peak areas were processed using MassHunter workstation (Agilent). Microsoft Excel and GraphPad Prism software were used for statistical analysis.

## Amino acid extraction and LC/MS/MS analysis

For amino acid analysis, five lymph glands from wandering 3$^{rd}$ instar larvae were taken and TCA (trichloroacetic acid) precipitation method was utilized. Briefly, lymph glands were dissected in 200µl of 1X PBS (Gibco) and 40µl of TCA (100% TCA soln.) was added. Samples were homogenized and incubated in ice for 10 minutes. After this, samples were centrifuged at 13000 RPM for 10 minutes at 4 degrees. Supernatant was transferred to fresh Eppendorf tubes and stored at -80 degrees until analysis. LC/MS/MS based analysis for amino acids was done utilizing HILIC chromatography on Agilent UPLC-triple quadrupole 6470 system.

## 13C-isotopic labelling and stable isotope tracer analysis for metabolite measurements

For isotopomer tracer analysis, wandering 3$^{rd}$ instar larvae were washed twice in PBS and lymph glands were dissected. Larval lymph glands were incubated in 10mM of U$^{13}$C-Pyruvate (Cambridge Isotope Laboratories, CLM-2440) in 1X PBS (Gibco) for 30 min. at room temperature. Samples were quickly rinsed in LC/MS grade water and processed for metabolite extraction and derivatization for metabolite analysis.

For LC/MS based steady-state and flux analysis five lymph glands per replicate (n) were taken and each experiment was repeated a minimum of three times (N). Q1/Q3 and R$_T$ values are provided in S1 Table.

## Statistical analyses

All statistical analyses and quantifications were performed using GraphPad Prism 10 and Microsoft Excel 2016. Mann-Whitney test is employed for statistical significance for all the experiments except the LC/MS based analysis, where t-test is applied to determine significance among the independent experimental repeats. In box and whisker plots, the horizontal line indicates the median, whiskers indicate the minimum and maximum values, and the box indicates the lower and upper quartiles. Statistical analyses and sample size were approached with the same level of rigor as done in our previous study [25]. Plots, test applied and sample size is mentioned separately in the figure legends section.

The schematic diagrams in each figure are drawn using Adobe Illustrator 2025 and figure panels are made using Adobe Photoshop 2025.

## Supporting information

**S1 Fig. GABA catabolic pathway in *Drosophila* blood-progenitor cells control their GSH levels.** RF is regular food, and GSH food and succinate food is GSH and succinate supplemented food respectively. Data is presented as median plots (****p<0.0001). Mann-Whitney test is applied for **D**. In **D**, 'n' is total number of lymph gland lobes analysed and is represented by a green dot. Scale bar: 20µm. DNA is stained with DAPI in blue, dome marks the progenitor cells in green. Comparisons for significance are done with their respective control. White border demarcates the lymph gland lobe and yellow border marks the dome positive area towards the left side. **(A-C")** Representative images showing GSH levels in the lymph gland progenitor cells shown by merge of dome+ (green), DNA (blue) and GSH (red) from *domeMeso-Gal4,UAS-GFP*/+ genetic backgrounds upon different washing conditions. **(A-A")** GSH (red) levels in the progenitor cells upon stringent washing with 0.3% triton x-100 show a reduced expression with distinction of GSH levels in the medullary and cortical zone and **(B-B")** GSH (red) levels in the progenitor cells upon mild washing with 0.1% tween-20 show an elevated expression, **(C-C")** supplementing control animals with GSH (GSH, *domeMeso-Gal4,UAS-GFP*/+) and utilization of stringent washing protocol leads to a significant increase in GSH levels in the progenitor cells. For comparison, refer to control (**A-A"**, Regular Food). For quantifications of GSH levels in regular food and GSH supplementation, refer to **D**. **(D)** Quantification of blood-progenitor GSH levels in *domeMeso>GFP*/+ (control, Regular Food, n=28), and *domeMeso>GFP*/+ (GSH Food, n=27, p<0.0001). **(E-G)** Representative images showing ROS levels in the lymph gland progenitor cells shown by merge of dome+ (green) and ROS (red) from different genetic backgrounds. Control (*domeMeso-Gal4,UAS-GFP*/+) showing elevated **(E)** ROS (stained with DHE, red) levels in the lymph gland progenitor cells, expressing **(F)** *Gat^RNAi* (*domeMeso-Gal4,UAS-GFP;UAS-Gat^RNAi*) and **(G)** *Ssadh^RNAi* (*domeMeso-Gal4,UAS-GFP;UAS-Ssadh^RNAi*) leads to an increase in blood-progenitor ROS levels as compared to control **(E)**. **(H-J')** Representative images showing GSH levels in the lymph gland progenitor cells shown by merge of dome+ (green), DNA (blue) and GSH (red) from different genetic backgrounds. **(H, H')** Control (*domeMeso-Gal4,UAS-GFP*/+) showing elevated **(E)** GSH levels in the lymph gland progenitor cells and, expressing **(I, I')** *Gat^RNAi* (*domeMeso-Gal4,UAS-GFP;UAS-Gat^RNAi*) and **(J, J')** *Ssadh^RNAi* (*domeMeso-Gal4,UAS-GFP;UAS-Ssadh^RNAi*) leads to a decrease in blood-progenitor GSH levels as compared to control **(H, H')**. **(K-N')** Succinate supplementation to **(K, M, M')** *Gat^RNAi* (*domeMeso-Gal4,UAS-GFP;UAS-Gat^RNAi*) and **(L, N, N')** *Ssadh^RNAi* (*domeMeso-Gal4,UAS-GFP;UAS-Ssadh^RNAi*) leads to recovery of blood-progenitor **(K, L)** ROS and **(M-N')** GSH defect. For comparison, refer to *Gat^RNAi* (**F**, ROS and **I, I'**, GSH) and *Ssadh^RNAi* (**G**, ROS and **J, J'**, GSH) raised on regular food.
(TIF)

**S2 Fig. Regulation of glutathione synthesis by GABA catabolic pathway.** RF is regular, SF is succinate, NAC is N-acetylcysteine and Met is methionine supplemented food. Data is presented as median plots (*p<0.05; **p<0.01; ***p<0.001; ****p<0.0001 and ns is non-significant). T-test is applied for **U, V** and Mann-Whitney test is applied for **W, X**. In **U, V** 'n' is sample size and 'N' is number of experimental repeats and shown by black dot. In **W, X** 'n' is total number of lymph gland lobes analysed and is represented by a green dot. Scale bar: 20µm. DNA is stained with DAPI in blue, dome marks the progenitor cells in green. Comparisons for significance are done with control and with respective genetic conditions for rescue combinations (red bars), which are indicated by horizontal lines drawn above the box plots. White border demarcates the lymph gland lobe and yellow border marks the dome positive area towards the left side. **(A-C)** Representative images showing glutamate levels in lymph gland progenitor cells (area marked within the yellow dotted line) with merge of dome+ (green), DNA (blue) and glutamate (Glu, red) across different genetic backgrounds. **(A)** Control (*domeMeso-Gal4,UAS-GFP*/+) lymph gland showing relatively uniform glutamate levels across all cells of the tissue, including the progenitor-cells (area within the yellow dotted line). Expressing **(B)** *Gat^RNAi* (*domeMeso-Gal4,UAS-GFP;UAS-Gat^RNAi*) and **(C)** *Ssadh^RNAi* (*domeMeso-Gal4,UAS-GFP;UAS-Ssadh^RNAi*) in the progenitor cells does not affect their glutamate levels in comparison to control **(A)**. **(D-F)** Representative images showing cysteine (Cys, red) levels in

lymph gland progenitor cells (area marked within the yellow dotted line) with merge of dome+ (green), DNA (blue) and cysteine (red) from different genetic backgrounds. **(D)** Control (RF, *domeMeso-Gal4,UAS-GFP/+*) lymph gland showing relatively uniform cysteine levels in all cells of the lymph gland including progenitor-cells (area demarcated within the yellow border). Expressing **(E)** *Gat^RNAi^* (RF, *domeMeso-Gal4,UAS-GFP;UAS-Gat^RNAi^*) and **(F)** *Ssadh^RNAi^* (RF, *domeMeso-Gal4,UAS-GFP;UAS-Ssadh^RNAi^*) in the progenitor cells leads to reduction in cysteine levels as compared to control **(D)**. **(G-L)** Representative images showing GSH (red) levels in lymph gland progenitor cells (area marked within the yellow dotted line) with merge of dome+ (green), DNA (blue) and GSH (red) from different genetic backgrounds. **(G)** Control (RF, *domeMeso-Gal4,UAS-GFP/+*) lymph gland showing GSH levels. While expressing **(H)** *Gat^RNAi^* (RF, *domeMeso-Gal4,UAS-GFP;UAS-Gat^RNAi^*) and **(I)** *Ssadh^RNAi^* (RF, *domeMeso-Gal4,UAS-GFP;UAS-Ssadh^RNAi^*) in the progenitor cells leads to reduction in GSH levels, supplementing these genetic conditions with **(J, K)** N-acetylcysteine (NAC), restores GSH levels in both **(J)** *Gat^RNAi^* (NAC, *domeMeso-Gal4,UAS-GFP;UAS-Gat^RNAi^*) and **(K)** *Ssadh^RNAi^* (NAC, *domeMeso-Gal4,UAS-GFP;UAS-Ssadh^RNAi^*) to levels almost comparable to control shown in **(G)**. **(L)** Methionine supplementation to *Gat^RNAi^* (Met, *domeMeso-Gal4,UAS-GFP;UAS-Gat^RNAi^*) does not recover GSH levels. For comparison also refer to *Gat^RNAi^* **(J)** and *Ssadh^RNAi^* **(K)** raised on regular food (RF). **(M-O)** Succinate supplementation to **(M)** *Gat^RNAi^* (SF, *domeMeso-Gal4,UAS-GFP;UAS-Gat^RNAi^*) and **(N)** *Ssadh^RNAi^* (SF, *domeMeso-Gal4,UAS-GFP;UAS-Ssadh^RNAi^*) restores blood-progenitor cysteine levels. For comparison refer to **(E)** *Gat^RNAi^* and **(F)** *Ssadh^RNAi^* animals raised on regular food (RF) and control **(D)**. **(O)** Progenitor specific expression of *Pdha^RNAi^* in *Gat^RNAi^* (*domeMeso-Gal4,UAS-GFP; UAS-Pdha^RNAi^;UAS-Gat^RNAi^*) leads to a recovery of blood-progenitor cysteine levels. Compare with **(E)** *Gat^RNAi^*. **(P, Q')** Expressing **(P, P')** *Pdk^RNAi^* (*domeMeso-Gal4,UAS-GFP;UAS-Pdk^RNAi^*) and **(Q, Q')** *Pdha^RNAi^* (*domeMeso-Gal4,UAS-GFP;UAS-Pdha^RNAi^*) in the progenitor cells, does not show any change in their cysteine levels. Compare to control **(D)**. For quantifications, refer to **W**. **(R-T)** Expressing **(R)** *Pdha^RNAi^* in *Gat^RNAi^* (*domeMeso-Gal4,UAS-GFP; UAS-Pdha^RNAi^;UAS-Gat^RNAi^*) leads to a recovery of blood-progenitor GSH levels. Compare with **(H)** *Gat^RNAi^*. Expressing **(S)** *Pdk^RNAi^* (*domeMeso-Gal4,UAS-GFP;UAS-Pdk^RNAi^*) and **(T)** *Pdha^RNAi^* (*domeMeso-Gal4,UAS-GFP;UAS-Pdha^RNAi^*) in the progenitor cells, does not reveal any dramatic change in GSH levels. Compare to control **(G)**. **(U)** Relative steady state levels (fold change, f.c.) of glutamate in *domeMeso>GFP/+* (control, N=7, n=23), *domeMeso>GFP/Gat^RNAi^* (N=4, n=14, p=0.7283) and *domeMeso>GFP/Ssadh^RNAi^* (N=5, n=17, p=0.0920). **(V)** Relative steady state levels (fold change, f.c.) of glycine in *domeMeso>GFP/+* (control, N=7, n=23), *domeMeso>GFP/Gat^RNAi^* (N=4, n=14, p=0.3291) and *domeMeso>GFP/Ssadh^RNAi^* (N=5, n=17, p=0.0820). **(W)** Quantification of blood-progenitor cysteine levels (fold change, f.c.) in *domeMeso>GFP/+* (control, n=50), *domeMeso>GFP/Gat^RNAi^* (n=28, p=0.0077), *domeMeso>GFP/Pdha^RNAi^;Gat^RNAi^* (n=49, p=0.0005 in comparison to *Gat^RNAi^*), *domeMeso>GFP/Pdk^RNAi^* (n=26, p=0.3030) and *domeMeso>GFP/Pdha^RNAi^* (n=26, p=0.5386). **(X)** Quantification of blood-progenitor GSH levels (fold change, f.c.) in *domeMeso>GFP/+* (control, n=59), *domeMeso>GFP/Gat^RNAi^* (n=28, p<0.0001), *domeMeso>GFP/Pdha^RNAi^;Gat^RNAi^* (n=41, p=0.0001 in comparison to *Gat^RNAi^*), *domeMeso>GFP/Pdk^RNAi^* (n=41, p=0.8670) and *domeMeso>GFP/Pdha^RNAi^* (n=51, p=0.0301).
(TIF)

**S3 Fig. GABA controls PDH activity to regulate blood-progenitor GSH synthesis.** Data is presented as median plots (****p<0.0001). Mann-Whitney test is applied for **M, N** and 'n' is total number of lymph gland lobes analysed and is represented by a green dot. Scale bar: 20µm. DNA is stained with DAPI in blue, dome marks the progenitor cells in green. Comparisons for significance are done with their respective control. White border demarcates the lymph gland lobe and yellow border marks the dome positive area towards the left side. **(A-C')** Representative images showing PDH (red) levels in lymph gland progenitor cells (area marked within the yellow dotted line) with merge of dome+ (green), DNA (blue) and PDH (red) from different genetic backgrounds. **(A, A')** Control (*domeMeso-Gal4,UAS-GFP/+*) lymph gland showing PDH levels, expressing **(B, B')** *Pdha^RNAi^* (*domeMeso-Gal4,UAS-GFP;UAS-Pdha^RNAi^*) and **(C, C')** *Pdha^RNAi;^Gat^RNAi^* (*domeMeso-Gal4,UAS-GFP; Pdha^RNAi;^Gat^RNAi^*) in the progenitor cells leads to reduction in PDH levels. For quantifications, refer to **M**. **(D-F')** Representative images showing Gat (red) levels in lymph gland progenitor cells (area marked within the

yellow dotted line) with merge of dome+ (green), DNA (blue) and Gat (red) from different genetic backgrounds. **(D, D')** Control (*domeMeso-Gal4,UAS-GFP/+*) lymph gland showing Gat levels, expressing **(E, E')** *Gat^RNAi* (*domeMeso-Gal4,UAS-GFP;UAS-Gat^RNAi*) and **(F, F')** *Pdha^RNAi;Gat^RNAi* (*domeMeso-Gal4,UAS-GFP; Pdha^RNAi;Gat^RNAi*) in the progenitor cells leads to reduction in Gat levels. For quantifications, refer to **N**. **(G-I')** Representative images showing PDH (red) levels in lymph gland progenitor cells (area marked within the yellow dotted line) with merge of dome+ (green), DNA (blue) and PDH (red) from different genetic backgrounds. **(G, G')** Control (*domeMeso-Gal4,UAS-GFP/+*) lymph gland showing PDH levels, expressing **(H, H')** *Pdha^RNAi* (*domeMeso-Gal4,UAS-GFP;UAS-Pdha^RNAi*) leads to reduction in PDH levels and expressing **(I, I')** *Pdk^RNAi* (*domeMeso-Gal4,UAS-GFP; Pdk^RNAi*) in the progenitor cells does not show any change in PDH levels. **(J-L')** Representative images showing pPDH (red) levels in lymph gland progenitor cells (area marked within the yellow dotted line) with merge of dome+ (green), DNA (blue) and pPDH (red) from different genetic backgrounds. **(J, J')** Control (*domeMeso-Gal4,UAS-GFP/+*) lymph gland showing pPDH levels, expressing **(K, K')** *Pdha^RNAi* (*domeMeso-Gal4,UAS-GFP;UAS-Pdha^RNAi*) and **(L, L')** *Pdk^RNAi* (*domeMeso-Gal4,UAS-GFP; Pdk^RNAi*) in the progenitor cells leads to reduction in pPDH levels. **(M)** Quantification of blood-progenitor PDH levels (fold change, f.c.) in *domeMeso>GFP/+* (control, n = 55), *domeMeso>GFP/Pdha^RNAi* (n = 58, p < 0.0001) and *domeMeso>GFP/Pdha^RNAi;Gat^RNAi* (n = 54, p < 0.0001). **(N)** Quantification of blood-progenitor Gat levels (fold change, f.c.) in *domeMeso>GFP/+* (control, n = 32), *domeMeso>GFP/Gat^RNAi* (n = 45, p < 0.0001) and *domeMeso>GFP/Pdha^RNAi;Gat^RNAi* (n = 35, p < 0.0001).
(TIF)

**S4 Fig. GABA catabolism controls progenitor GSH generation via regulating *de novo* serine synthesis.** RF is regular food, Ser is serine and OAA is oxaloacetate supplemented food. Scale bar: 20μm. DNA is stained with DAPI in blue, dome marks the progenitor cells in green. White border demarcates the lymph gland lobe and yellow border marks the dome positive area towards the left side. **(A-C)** Representative images showing cysteine (Cys, red) levels in lymph gland progenitor cells (area marked within the yellow dotted line) with merge of dome+ (green), DNA (blue) and cysteine (red) from different genetic backgrounds. **(A)** Control (RF, *domeMeso-Gal4,UAS-GFP/+*) lymph gland showing relatively uniform cysteine levels in all cells of the lymph gland including progenitor-cells (area demarcated within the yellow border). While, expressing **(B)** *Gat^RNAi* (RF, *domeMeso-Gal4,UAS-GFP;UAS-Gat^RNAi*) in the progenitor cells leads to reduction in cysteine levels as compared to control **(A)**, supplementing this genetic condition with **(C)** serine (Ser, *domeMeso-Gal4,UAS-GFP;UAS-Gat^RNAi*) recovers cysteine levels almost comparable to control **(A)**. For comparison, also refer to **(B)** *Gat^RNAi* raised on regular food (RF). **(D-F)** Representative images showing GSH (red) levels in lymph gland progenitor cells (area marked within the yellow dotted line) with merge of dome+ (green), DNA (blue) and GSH (red) from different genetic backgrounds. **(D)** Control (RF, *domeMeso-Gal4,UAS-GFP/+*) lymph gland showing GSH levels in lymph gland progenitor-cells (area demarcated within the yellow border). While, expressing **(E)** *Gat^RNAi* (RF, *domeMeso-Gal4,UAS-GFP;UAS-Gat^RNAi*) in the progenitor cells leads to reduction in GSH levels as compared to control **(D)**, supplementing this genetic condition with **(F)** serine (Ser, *domeMeso-Gal4,UAS-GFP;UAS-Gat^RNAi*) recovers GSH levels almost comparable to control **(D)**. For comparison, also refer to **(E)** *Gat^RNAi* raised on regular food (RF). **(G-I)** Representative images showing ROS levels (area marked within the yellow dotted line) with merge of dome+ (green) and ROS (red) from different genetic backgrounds. **(G)** control (RF, *domeMeso-Gal4,UAS-GFP/+*) lymph gland showing ROS levels in lymph gland progenitor-cells (area demarcated within the yellow border). While, expressing **(H)** *Gat^RNAi* (RF, *domeMeso-Gal4,UAS-GFP;UAS-Gat^RNAi*) leads to increase in progenitor ROS as compared to control **(G)**, supplementing this genetic condition with **(I)** serine (Ser, *domeMeso-Gal4,UAS-GFP;UAS-Gat^RNAi*) recovers the increased ROS levels. For comparison, also refer to **(H)** *Gat^RNAi* raised on regular food (RF). **(J-N)** Representative images showing GSH (red) levels in lymph gland progenitor cells with merge of dome+ (green), DNA (blue) and GSH (red) from different genetic backgrounds. **(J)** Control (RF, *domeMeso-Gal4,UAS-GFP/+*) lymph gland showing GSH levels in progenitor-cells of the lymph gland. Expressing **(K)** *Gat^RNAi* (RF, *domeMeso-Gal4,UAS-GFP;UAS-Gat^RNAi*) in the progenitor cells leads to reduction in GSH levels as compared to control **(J)**, supplementing *Gat^RNAi* with **(L)** oxaloacetate (OAA), restores GSH levels in *Gat^RNAi* (OAA,

*domeMeso-Gal4,UAS-GFP;UAS-Gat^{RNAi}*), **(M)** progenitor specific expression of *Cs^{RNAi}* in *Gat^{RNAi}* (*domeMeso-Gal4,UAS-GFP; UAS-Cs^{RNAi};UAS-Gat^{RNAi}*) leads to a recovery of blood-progenitor GSH levels, compare with **(K)** GSH in *Gat^{RNAi}* and **(N)** progenitor-specific expression of *Cs^{RNAi}* (*domeMeso-Gal4,UAS-GFP; UAS-Cs^{RNAi}*) leads to increase in GSH levels as compared to control **(J)**. **(O-S)** Representative images showing cysteine (Cys, red) levels in lymph gland progenitor cells with merge of dome+ (green), DNA (blue) and Cys (red) from different genetic backgrounds. **(O)** Control (RF, *domeMeso-Gal4,UAS-GFP/+*) lymph gland showing Cys levels in progenitor-cells of the lymph gland. Expressing **(P)** *Gat^{RNAi}* (RF, *domeMeso-Gal4,UAS-GFP;UAS-Gat^{RNAi}*) in the progenitor cells leads to reduction in Cys levels as compared to control **(O)**, supplementing *Gat^{RNAi}* with **(Q)** oxaloacetate (OAA), restores Cys levels in *Gat^{RNAi}* (OAA, *domeMeso-Gal4,UAS-GFP;UAS-Gat^{RNAi}*), **(R)** progenitor specific expression of *Cs^{RNAi}* in *Gat^{RNAi}* (*domeMeso-Gal4,UAS-GFP; UAS-Cs^{RNAi};UAS-Gat^{RNAi}*) leads to a recovery of blood-progenitor Cys levels, compare with **(P)** Cys in *Gat^{RNAi}* and **(S)** progenitor-specific expression of *Cs^{RNAi}* (*domeMeso-Gal4,UAS-GFP; UAS-Cs^{RNAi}*) leads to increase in Cys levels as compared to control **(O)**.
(TIF)

**S5 Fig. Oxaloacetate (OAA) flux into gluconeogenic arm and serine synthesis regulate GSH homeostasis.** RF is regular food and NAC is N-acetylcysteine supplemented food. Data is presented as median plots (****$p < 0.0001$ and ns is non-significant). Mann-Whitney test is applied for **P-R**. In **P-R**, 'n' is total number of lymph gland lobes analysed and is represented by a green dot. Scale bar: 20μm. DNA is stained with DAPI in blue. Comparisons for significance are done with their respective control. White border demarcates the lymph gland lobe and yellow border marks the dome positive area towards the left side. **(A-B')** Representative images showing glutathione (GSH, red) levels in lymph gland progenitor cells (area marked within the yellow dotted line) and with merge of dome+ (green), DNA (blue) and GSH (red) from different genetic backgrounds. In comparison to **(A, A')** control (*domeMeso-Gal4,UAS-GFP/+*) lymph gland, expressing **(B, B')** *Pepck1^{RNAi}* (*domeMeso-Gal4,UAS-GFP;UAS-Pepck1^{RNAi}*) in the progenitor cells does not show any change in blood-progenitor GSH levels. For quantifications, refer to **P**. **(C-L)** Representative images showing glutathione (GSH, red) and ROS (red) levels in lymph gland progenitor cells with merge of dome+ (green), DNA (blue) and GSH/ROS (red) from different genetic backgrounds. **(C)** control (*domeMeso-Gal4,UAS-GFP/+*) lymph gland showing GSH levels, expressing **(D)** *Pepck2^{RNAi}* (*domeMeso-Gal4,UAS-GFP;UAS-Pepck2^{RNAi}*) leads to reduction in blood-progenitor GSH levels. **(E)** control (*domeMeso-Gal4,UAS-GFP/+*) lymph gland showing ROS levels, expressing **(F)** *Pepck2^{RNAi}* (*domeMeso-Gal4,UAS-GFP;UAS-Pepck2^{RNAi}*) leads to elevation of blood-progenitor ROS levels and similarly expressing **(G, H)** *Phgdh^{RNAi}* (*domeMeso-Gal4,UAS-GFP;UAS-Phgdh^{RNAi}*) leads to reduction in blood-progenitor **(G)** GSH levels and increase in **(H)** ROS levels. Compare to control **(C)** GSH and **(E)** ROS. NAC supplementation to **(I, K)** *Pepck2^{RNAi}* (NAC, *domeMeso-Gal4,UAS-GFP;UAS-Pepck2^{RNAi}*) and **(J, L)** *Phgdh^{RNAi}* (NAC, *domeMeso-Gal4,UAS-GFP;UAS-Phgdh^{RNAi}*) leads to recovery of blood progenitor **(I, J)** GSH and **(K, L)** ROS levels. For comparison refer to **(D, G)** GSH and **(F, H)** ROS in *Pepck2^{RNAi}* (RF, *domeMeso-Gal4,UAS-GFP;UAS-Pepck2^{RNAi}*) and *Phgdh^{RNAi}* (RF, *domeMeso-Gal4,UAS-GFP;UAS-Phgdh^{RNAi}*) respectively. **(M-O')** Representative images showing Cys (red) levels in lymph gland progenitor cells (area marked within the yellow dotted line) and with merge of dome+ (green), DNA (blue) and Cys (red) from different genetic backgrounds. In comparison to **(M, M')** control (*domeMeso-Gal4,UAS-GFP/+*) lymph gland, expressing **(N, N')** *Pepck2^{RNAi}* (*domeMeso-Gal4,UAS-GFP;UAS-Pepck2^{RNAi}*) leads to reduction in blood-progenitor Cys levels and expressing **(O, O')** *Phgdh^{RNAi}* (*domeMeso-Gal4,UAS-GFP;UAS-Phgdh^{RNAi}*) leads to an increase in blood-progenitor Cys levels. For quantifications, refer to **R**. **(P)** Quantification of blood-progenitor GSH levels (fold change, f.c.) in *domeMeso>GFP/+* (control, n = 43) and *domeMeso>GFP/Pepck1^{RNAi}* (n = 39, p = 0.0792). **(Q)** Quantification of lymph gland area in *domeMeso>GFP/+* (control, n = 43) and *domeMeso>GFP/Pepck1^{RNAi}* (n = 39, p = 0.1915). **(R)** Quantification of blood-progenitor Cys levels (fold change, f.c.) in *domeMeso>GFP/+* (control, n = 46), *domeMeso>GFP/Pepck2^{RNAi}* (n = 33, p < 0.0001) and *domeMeso>GFP/Phgdh^{RNAi}* (n = 31, p < 0.0001).
(TIF)

**S6 Fig. Oxaloacetate (OAA) flux into gluconeogenic arm and serine synthesis regulate GSH homeostasis.** RF is regular food, Ser is serine and Gly is glycine supplemented food. Data is presented as median plots (*p<0.05; **p<0.01; ***p<0.001; ****p<0.0001 and ns is non-significant). Mann-Whitney test is applied for **M-P**. In **M-P**, 'n' is total number of lymph gland lobes analysed and is represented by a green dot. Scale bar: 20µm. DNA is stained with DAPI in blue, dome marks the progenitor cells in green. Comparisons for significance are done with control and with respective genetic conditions for rescue combinations (red bars), which are indicated by horizontal lines drawn above the box plots. White border demarcates the lymph gland lobe and yellow border marks the dome positive area towards the left side. **(A-F')** Representative images showing glutathione (GSH, red) levels in lymph gland progenitor cells (area marked within the yellow dotted line) and with merge of dome+ (green), DNA (blue) and GSH (red) from different genetic backgrounds. In comparison to **(A, A')** control (RF, *domeMeso-Gal4,UAS-GFP*/+) lymph gland, expressing **(B, B')** *Pepck2$^{RNAi}$* (RF, *domeMeso-Gal4,UAS-GFP;UAS-Pepck2$^{RNAi}$*) leads to reduction in blood-progenitor GSH levels as compared to control **(A, A')** and **(C, C')** serine supplementation in *Pepck2$^{RNAi}$* (Ser, *domeMeso-Gal4,UAS-GFP;UAS-Pepck2$^{RNAi}$*) leads to recovery of blood progenitor GSH levels as compared to **(B, B')** *Pepck2$^{RNAi}$* on RF, expressing **(D, D')** *Phgdh$^{RNAi}$* (RF, *domeMeso-Gal4,UAS-GFP;UAS-Phgdh$^{RNAi}$*) leads to reduction in blood-progenitor GSH levels as compared to control **(A, A')** and **(E, E')** serine supplementation in *Phgdh$^{RNAi}$* (Ser, *domeMeso-Gal4,UAS-GFP;UAS-Phgdh$^{RNAi}$*) recovers blood-progenitor GSH levels as compared to **(D, D')** *Phgdh$^{RNAi}$* on RF. For quantifications, refer to **M**. **(F-J')** Representative images showing Cys (red) levels in lymph gland progenitor cells (area marked within the yellow dotted line) and with merge of dome+ (green), DNA (blue) and Cys (red) from different genetic backgrounds. In comparison to **(F, F')** control (RF, *domeMeso-Gal4,UAS-GFP*/+) lymph gland, expressing **(G, G')** *Pepck2$^{RNAi}$* (RF, *domeMeso-Gal4,UAS-GFP;UAS-Pepck2$^{RNAi}$*) leads to reduction in blood-progenitor Cys levels as compared to control **(F, F')** and **(H, H')** serine supplementation in *Pepck2$^{RNAi}$* (Ser, *domeMeso-Gal4,UAS-GFP;UAS-Pepck2$^{RNAi}$*) leads to recovery of blood progenitor Cys levels as compared to *Pepck2$^{RNAi}$* on RF **(G, G')**, expressing **(I, I')** *Phgdh$^{RNAi}$* (RF, *domeMeso-Gal4,UAS-GFP;UAS-Phgdh$^{RNAi}$*) show an increase in blood-progenitor Cys levels as compared to control **(F, F')** and **(J, J')** serine supplementation in *Phgdh$^{RNAi}$* (Ser, *domeMeso-Gal4,UAS-GFP;UAS-Phgdh$^{RNAi}$*) leads to further elevation of blood progenitor Cys levels as compared to **(I, I')** *Phgdh$^{RNAi}$* on RF. For quantifications, refer to **N**. **(K-L')** Glycine supplementation to *Phgdh$^{RNAi}$* (Gly, *domeMeso-Gal4,UAS-GFP;UAS-Phgdh$^{RNAi}$*) leads to recovery of blood progenitor **(K, K')** GSH levels as compared to **(D, D')** *Phgdh$^{RNAi}$* on RF and further elevation of **(L, L')** cysteine levels as compared to **(I, I')**-*Phgdh$^{RNAi}$* on RF. For quantifications, refer to **M** and **N**. **(M)** Quantification of blood-progenitor GSH levels (fold change, f.c.) in *domeMeso>GFP*/+ (control, RF, n=54), *domeMeso>GFP/Pepck2$^{RNAi}$* (RF, n=29, p<0.0001), *domeMeso>GFP/Pepck2$^{RNAi}$* (Ser, n=33, p<0.0001 in comparison to *Pepck2$^{RNAi}$*, RF), *domeMeso>GFP/Phgdh$^{RNAi}$* (RF, n=28, p<0.0001), *domeMeso>GFP/Phgdh$^{RNAi}$* (Ser, n=15, p<0.0001 in comparison to *Phgdh$^{RNAi}$*, RF) and *domeMeso>GFP/Phgdh$^{RNAi}$* (Gly, n=26, p<0.0001 in comparison to *Phgdh$^{RNAi}$*, RF). **(N)** Quantification of blood-progenitor Cys levels (fold change, f.c.) in *domeMeso>GFP*/+ (control, RF, n=48), *domeMeso>GFP/Pepck2$^{RNAi}$* (RF, n=35, p=0.0001), *domeMeso>GFP/Pepck2$^{RNAi}$* (Ser, n=34, p<0.0001 in comparison to *Pepck2$^{RNAi}$*, RF), *domeMeso>GFP/Phgdh$^{RNAi}$* (RF, n=9, p<0.0001), *domeMeso>GFP/Phgdh$^{RNAi}$* (Ser, n=5, p=0.1119 in comparison to *Phgdh$^{RNAi}$*, RF) and *domeMeso>GFP/Phgdh$^{RNAi}$* (Gly, n=8, p=0.3213 in comparison to *Phgdh$^{RNAi}$*, RF). **(O)** Quantification of lymph gland area in *domeMeso>GFP*/+ (control, RF, n=71), *domeMeso>GFP/Pepck2$^{RNAi}$* (RF, n=24, p<0.0001), *domeMeso>GFP/Pepck2$^{RNAi}$* (Ser, n=24, p=0.0459 in comparison to *Pepck2$^{RNAi}$*, RF), *domeMeso>GFP/Phgdh$^{RNAi}$* (RF, n=40, p<0.0001), *domeMeso>GFP/Phgdh$^{RNAi}$* (Ser, n=23, p=0.0027 in comparison to *Phgdh$^{RNAi}$*, RF) and *domeMeso>GFP/Phgdh$^{RNAi}$* (Gly, n=34, p=0.6390 in comparison to *Phgdh$^{RNAi}$*, RF). **(P)** Quantification of GFP intensity in *domeMeso>GFP*/+ (control, n=33), *domeMeso>GFP/Phgdh$^{RNAi}$* (n=37, p=0.6150), *domeMeso>GFP/Pepck2$^{RNAi}$* (n=31, p=0.4306), *domeMeso>GFP/Pepck1$^{RNAi}$* (n=24, p=0.8537), *domeMeso>GFP/Ssadh$^{RNAi}$* (n=16, p=0.4791), *domeMeso>GFP/Pdk$^{RNAi}$* (n=16, p=0.7437), *domeMeso>GFP/Pdha$^{RNAi}$* (n=14, p=0.0966), *domeMeso>GFP/Cs$^{RNAi}$* (n=9, p=0.8330), *domeMeso>GFP/Gat$^{RNAi}$* (n=37, p=0.1047), *domeMeso>GFP/Pdha$^{RNAi}$;Gat$^{RNAi}$* (n=22, p=0.1249 and

p > 0.9999 in comparison to *Gat^RNAi*) and *domeMeso>GFP/Cs^RNAi;Gat^RNAi* (n = 8, p = 0.8592 and p = 0.6515 in comparison to *Gat^RNAi*).
(TIF)

**S1 Table. Q1/Q3 and R$_T$ values for the metabolites detected by LC/MS/MS.**
(DOCX)

**S1 Data. Excel file containing the quantitative data used for graphs presented in Figs 1–5 and S1–S6.**
(XLSX)

## Acknowledgments

We thank the Bloomington *Drosophila* Stock Center (BDSC) and Vienna *Drosophila* Resource Center (VDRC) for fly stocks. We acknowledge the support of Central Imaging & Flow Cytometry Facility (CIFF) and the fly facility at National Centre for Biological Sciences (NCBS), Centre for Cellular and Molecular Platforms (C-CAMP). We thank metabolomics facility at MPF, Bindley Bioscience Center for help with metabolomics experiments and Dr. Vikki Weake, Purdue University for providing the space for *Drosophila* related work. We thank Patrick Jouandin and Norbert Perrimon for their suggestions and feedback on metabolomics experiments. Owing to space limitations, we apologize to our colleagues whose work is not cited.

## Author contributions

**Conceptualization:** Manisha Goyal, Tina Mukherjee.

**Data curation:** Manisha Goyal, Sakshi Tiwari, Jagriti Arora.

**Formal analysis:** Manisha Goyal.

**Funding acquisition:** Ramaswamy Subramanian, Tina Mukherjee.

**Investigation:** Manisha Goyal, Sakshi Tiwari, Jagriti Arora.

**Methodology:** Manisha Goyal, Sakshi Tiwari, Jagriti Arora, Bruce Cooper.

**Project administration:** Tina Mukherjee.

**Resources:** Ramaswamy Subramanian, Tina Mukherjee.

**Software:** Tina Mukherjee.

**Supervision:** Bruce Cooper, Ramaswamy Subramanian, Tina Mukherjee.

**Validation:** Manisha Goyal, Sakshi Tiwari, Jagriti Arora.

**Visualization:** Manisha Goyal.

**Writing – original draft:** Manisha Goyal, Tina Mukherjee.

**Writing – review & editing:** Manisha Goyal, Ramaswamy Subramanian, Tina Mukherjee.

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
