## [Decision Letter · Decision Letter 0]

15 Apr 2025

PGENETICS-D-25-00122

Antioxidant role of the GABA shunt in regulating redox balance in blood progenitors during Drosophila hematopoiesis

PLOS Genetics

Dear Dr. Mukherjee,

Thank you for submitting your manuscript to PLOS Genetics. After careful consideration of the Revviwers comments, we feel that it has merit but does not fully meet PLOS Genetics's publication criteria as it currently stands. Therefore, we invite you to submit a revised version of the manuscript that addresses the points raised during the review process.

Please submit your revised manuscript within 60 days Jun 14 2025 11:59PM. If you will need more time than this to complete your revisions, please reply to this message or contact the journal office at plosgenetics@plos.org. Please include the following items when submitting your revised manuscript:

We look forward to receiving your revised manuscript.

Kind regards,

Lolitika Mandal, Ph.D

Academic Editor

PLOS Genetics

Pablo Wappner

Section Editor

PLOS Genetics

Aimée Dudley

Editor-in-Chief

PLOS Genetics

Anne Goriely

Editor-in-Chief

PLOS Genetics

**Journal Requirements:**

1) Please provide an Author Summary. This should appear in your manuscript between the Abstract (if applicable) and the Introduction, and should be 150-200 words long. The aim should be to make your findings accessible to a wide audience that includes both scientists and non-scientists. Sample summaries can be found on our website under Submission Guidelines:

https://journals.plos.org/plosgenetics/s/submission-guidelines#loc-parts-of-a-submission

3) We have noticed that you have uploaded Supporting Information files, but you have not included a complete list of legends. Please add a full list of legends for your Supporting Information files after the references list.

4) We notice that your supplementary figures are uploaded with the file type 'Figure'. Please amend the file type to 'Supporting Information'. Please ensure that each Supporting Information file has a legend listed in the manuscript after the references list.

Potential Copyright Issues:

i) Figures 1A, and 4F. Please confirm whether you drew the images / clip-art within the figure panels by hand. If you did not draw the images, please provide (a) a link to the source of the images or icons and their license / terms of use; or (b) written permission from the copyright holder to publish the images or icons under our CC BY 4.0 license. Alternatively, you may replace the images with open source alternatives. See these open source resources you may use to replace images / clip-art:

7) Please ensure that the funders and grant numbers match between the Financial Disclosure field and the Funding Information tab in your submission form. Note that the funders must be provided in the same order in both places as well. Currently, the order of the funders is different in both places.

**Reviewers' comments:**

Reviewer's Responses to Questions

Reviewer #1: The authors discovered that GSH is synthesized in blood progenitor cells in Drosophila lymph glands, and they investigated how GSH was generated in progenitor cells. Interestingly, antioxidant GSH was produced in cells producing ROS, which has been previously shown to be essential to maintaining the properties of progenitor cells. However, the authors did not address whether GSH production is crucial to maintaining ROS homeostasis or the properties of the blood progenitor cells. Thus, the paper describes possible pathways that generate GSH in the blood progenitor cells. There are problems with the interpretation of results as described below.

Major concerns:

1) Quantification of free amino acids: The authors quantified free amino acids based on the signal intensity of immunohistochemical staining. The method is not generally used, and I would like to know whether it is possible to fix free amino acids in tissues so they won't be washed away during the staining procedure. It would be helpful if the authors could provide any data or references to support the reliability of the methods.

2) RNAi experiments. The authors performed RNAi-mediated gene knockdown. The efficiency of knockdown depends on the constructs, and it is essential to know the extent of downregulation to interpret the results. The efficiency was also affected by the number of UAS transgenes expressed simultaneously. I understand that UAS-GFP was always included in all RNAi experiments to visualize the population of progenitor cells. Therefore, two or three UAS constructs were under the control of GAL4. Since the amount of GAL4 protein is limited, RNAi efficiency would be variable depending on the number of UAS constructs. Quantification of gene expression levels is necessary in knockdown experiments.

3) Line 211-216: The authors downregulated Pdha in Gat RNAi condition, assessed for cysteine and GSH levels, and concluded that downregulated Pdha suppressed the GABA component RNAi phenotype. However, as I mentioned above, it is also possible that the downregulation of Gat RNAi was inefficient because of the co-expression of two RNAi constructs. Therefore, the role of Pdha in GSH and cysteine levels remains unclear.

4) 221-234: The authors performed downregulation of Pdk or Pdha, and neither affected the GSH and cysteine levels. These data suggest two possibilities: 1) RNAi did not work efficiently. Verification of knockdown is necessary. 2) Both RNAi worked adequately, but neither is involved in regulating GSH and cysteine levels.

Minor comments:

Lines 133-139. should be described in the Discussion

Line 174-178. Progenitor-specific loss of Gat or Ssadh led to a significant reduction in overall cysteine levels (Fig. 2E-G, U and Fig. S2D-F)

Why was cysteine level reduced in differentiated cells that do not express Gat or Ssadh RNAi?

235-245 should be in Discussion.

Reviewer #2: This study by Goyal and colleagues examines the metabolic mechanisms regulating glutathione biosynthesis in the larval lymph gland. Previous studies from this lab demonstrated that GABA metabolism restricts PDH activity, thus regulating flux through the TCA cycle. Here the authors demonstrate that this function of GABA metabolism influences glutathione (GSH) levels by indirectly promoting serine biosynthesis.

There are two key observations in the study. First, increased pyruvate oxidation via PDH does not restore GSH synthesis. Second, the authors present data that support a model in which pyruvate is metabolized to oxaloacetate via pyruvate carboxylase – I applaud them for using stable-isotope tracing to advance their studies. Together, these observations hint at a model in which serine biosynthesis in the lymph gland is the product of gluconeogenesis.

Overall, I find the basic premises of the work quite exciting – the authors present data on GSH metabolism in the fly lymph gland that will no doubt be used as the foundation of developmental metabolism studies for years to come - I really like this study. Below I suggest a few experiments to solidify the model prior to publication.

1. Their model suggests that gluconeogenesis is regulating GSH production. I’d like to see knockdown of Pepck expression using RNAi, as this should similarly induce a GSH depletion.

2. Similarly, disruption of the serine biosynthetic pathway using RNAi in the lymph gland should result in diminished GSH production.

3. Finally, the experiments described above should include a rescue, demonstrating that cysteine feeding restores GSH levels following Pepck and Phgdh RNAi.

4. Are cysteine levels low because of decreased synthesis or because of elevated cystine formation, which is formed from two cysteine models in response to increased ROS. I don’t feel this experiment is absolutely necessary, but if the authors already have metabolomics data, this should be an easy comment to address.

5. Please cite Flybase. This can be done in the methods using a statement such as “Flybase was used throughout the study (ref).” Such citations are key to demonstrate the importance of Flybase to US funding agencies.

In addition, I have a few minor comments:

Line 26 - GSH is undefined in the abstract. Please define the abbreviation or simply use the term glutathione in the abstract.

Line 64-65 – use of the word “cues” twice in the same sentence.

Line 67 – delete the word “any” before “aberrant”. This is redundant.

Line 133 – missing citation after “reported in literature”. Also, I think this should be reworded as “reported in the literature.”

Line 182-183 – This sentence is missing either words or commas, I’m not sure which.

Reviewer #3: In this work, the authors build upon their previous findings on GABA catabolism controlling TCA cycle activity (and the amount of ROS produced) in the Drosophila larval hematopoietic organ, the lymph gland. Here, the authors show that the suppression of TCA cycle activity via GABA catabolism allows for the production of glutathione, a tripeptide with an important role in ROS scavenging, via cysteine synthesis. The results give further insight into the mechanisms of maintaining a balance in the amount of ROS in progenitor cells. The experiments presented in the manuscript are generally well executed and back up the conclusions made. Below I present my (minor) comments on the manuscript.

I find the results of isotope analysis (Figure 4) to be the least convincing. For example, in Fig 4B, the OAA 13C3 amount seems to be reduced in the Gat knockdown LGs, but then looking at Fig4B’, the difference looks very minor, and even more so for citrate in Fig4C’. I’m left to wonder how much can be said based on these results/differences. Could the authors please comment on this, and edit the conclusions from this part, if necessary?

The statistical test used (2-way ANOVA) may not be ideal when the response/dependent variable is not continuous, but a proportion/percentage in this case. In addition, why was 2-way ANOVA used? From what I understand, it seems there is only one explanatory variable (genotype) and no interaction terms? This could be clarified.

Figure 1. Quantification of the ROS levels could be added.

Throughout the text, using the word “mutant” when referring to RNAi-mediated silencing is slightly misleading and could be replaced, for example, with “knockdown”.

Line 141 “TCA cycle activity”?

Line 149 Sentence starting “Succinate, which is an end-product...” has a bit unclear structure and could be revised.

Line 182 The sentence starting “These data revealed…” is unclear and needs to be revised.

Line 222 Full name of Pdk should be given.

Line 236 “TCA cycle”, the word “cycle” missing also elsewhere

Line 292 “pyruvate” misspelled

Line 566 What is meant by “Drosophila are not limiting”?

**Have all data underlying the figures and results presented in the manuscript been provided?**

Reviewer #1: Yes

Reviewer #2: Yes

Reviewer #3: Yes

PLOS authors have the option to publish the peer review history of their article (what does this mean? ). If published, this will include your full peer review and any attached files.

**Do you want your identity to be public for this peer review?** For information about this choice, including consent withdrawal, please see our Privacy Policy .

Reviewer #1: No

Reviewer #2: **Yes: ** Jason Tennessen

Reviewer #3: No

**Figure resubmission:**
---

## [Decision Letter · Decision Letter 1]

8 Sep 2025

Dear Dr Mukherjee,

We are pleased to inform you that your manuscript entitled "Metabolic coupling of ROS generation and antioxidant synthesis by the GABA shunt pathway in myeloid-like blood progenitor cells of Drosophila" has been editorially accepted for publication in PLOS Genetics. Congratulations!

Yours sincerely,

Lolitika Mandal, Ph.D

Academic Editor

PLOS Genetics

Pablo Wappner

Section Editor

PLOS Genetics

Aimée Dudley

Editor-in-Chief

PLOS Genetics

Anne Goriely

Editor-in-Chief

PLOS Genetics

Comments from the reviewers (if applicable):

Reviewer #1:

Reviewer #2:

Reviewer #3:

Reviewer's Responses to Questions

**Comments to the Authors:**

Reviewer #1: The authors have adequately addressed my concerns in the revised version.

I have no further comments.

Reviewer #2: The authors have addressed my concerns - the additional RNAi experiments significantly enhance the study. Nice job.

Reviewer #3: The authors have done very thorough job in addressing all the points raised by me and the other reviewers. I believe this is now a strong piece of work and I recommend it to be published in PLOS Genetics.

**Have all data underlying the figures and results presented in the manuscript been provided?**

Reviewer #1: Yes

Reviewer #2: Yes

Reviewer #3: Yes

PLOS authors have the option to publish the peer review history of their article (what does this mean? ). If published, this will include your full peer review and any attached files.

**Do you want your identity to be public for this peer review?** For information about this choice, including consent withdrawal, please see our Privacy Policy .

Reviewer #1: No

Reviewer #2: **Yes: ** Jason M. Tennessen

Reviewer #3: No

**Data Deposition**

http://datadryad.org/submit?journalID=pgenetics&manu=PGENETICS-D-25-00122R1

**Press Queries**

---

## [Editor Report · Acceptance letter]

PGENETICS-D-25-00122R1

Metabolic coupling of ROS generation and antioxidant synthesis by the GABA shunt pathway in myeloid-like blood progenitor cells of Drosophila

Dear Dr Mukherjee,

We are pleased to inform you that your manuscript entitled "Metabolic coupling of ROS generation and antioxidant synthesis by the GABA shunt pathway in myeloid-like blood progenitor cells of Drosophila" has been formally accepted for publication in PLOS Genetics! Your manuscript is now with our production department and you will be notified of the publication date in due course.

With kind regards,

Anita Estes

PLOS Genetics

On behalf of:
